

# Inhomogeneous quenches in the transverse field Ising chain: scaling and front dynamics

**Márton Kormos**

BME-MTA Statistical Field Theory Research Group, Institute of Physics,
Budapest University of Technology and Economics, H-1111 Budapest, Hungary

⋆ kormos@eik.bme.hu

## Abstract

We investigate the non-equilibrium dynamics of the transverse field quantum Ising chain evolving from an inhomogeneous initial state given by joining two macroscopically different semi-infinite chains. We obtain integral expressions for all two-point correlation functions of the Jordan–Wigner Majorana fermions at any time and for any value of the transverse field. Using this result, we analytically compute the profiles of various physical observables in the space-time scaling limit and show that they can be obtained from a hydrodynamic picture based on ballistically propagating quasiparticles. Going beyond the hydrodynamic limit, we analyze the approach to the non-equilibrium steady state and find that the leading late time corrections display a lattice effect. We also study the fine structure of the propagating fronts which are found to be described by the Airy kernel and its derivatives. Near the front we observe the phenomenon of energy back-flow where the energy locally flows from the colder to the hotter region.


# 1 Introduction

The last decade witnessed an ever increasing interest in the out of equilibrium dynamics of quantum many-body systems [1, 2]. To a large extent this is due to the spectacular advances in experiments performed with cold atoms which makes it possible to study the coherent time evolution of quantum systems that can be, to a good precision, considered isolated from their environment. Lattice and continuum systems can be studied and their parameters such as their coupling strengths, the confining potential etc. can be changed in real time, opening the way to study a large variety of dynamical phenomena [3–5].

The mechanism of equilibration and thermalization has been in the focus of attention of many theoretical works. The main paradigm has been the so-called quantum quench [6], i.e. a sudden change of a system parameter and the subsequent out of equilibrium time evolution. It was soon realized that integrable systems are peculiar [7] because in general they cannot reach thermal equilibrium, even locally, due to the presence of an extensive number of local conserved quantities [8].

In most of these studies, the initial state was taken to be the ground state of a locally interacting Hamiltonian. To be able to study transport properties, however, one needs to consider either an open system with driving at its boundaries, or in the case of isolated systems, an inhomogeneous initial state. The simplest inhomogeneous setup is arguably provided by the so-called partitioning protocol or tensor product initial state. This means that two systems with macroscopically different properties (temperature, magnetization, etc.) are suddenly connected to each other.[1]

---

[1]Sometimes this is also referred to as a local quench, although this can be misleading as in general the initial state has a finite energy density.

Integrable systems behave in a special way also in this setup. Fourier's law does not hold in them: they can support currents even in the absence of a gradient and they can feature exotic transport properties. In many integrable systems the transport can be ballistic, however, this is not necessary [9–11]. In particular, the non-equilibrium steady state (NESS) forming after a long time in the partitioning protocol is typically a translationally invariant state which however supports a finite current. Many works focused on the non-equilibrium steady state in the two-temperature scenario, and in particular on the computation of the expectation value of the energy current in the NESS in the XX [12,13], XY [14,15], Ising [16], XXZ [17–19] spin chains, in conformal [20–24] and integrable [25,26] field theories. Fluctuation relations for the currents have also been derived [22,27], and various bounds for the ballistic currents were discussed [28]. Subsystem correlations [29–31] and higher dimensional models with similar setups have also been investigated [21,32,33].

Apart from the NESS, the details of the non-equilibrium time evolution and the space-time dependent profile of physical quantities is also of great interest. When the transport is ballistic, at large times these profiles are given by functions of the scaling variable $x/t$, i.e. they are constant along a "ray" defined by this ratio. Profiles of the magnetization, particle and energy density were studied in various systems, including the XX spin chain (free hopping fermions) [34–39], the XY and Ising spin chain [36,40–43], continuum free fermions [44], and conformal field theory [45,46]. Inhomogeneous initial states were studied extensively in the XXZ spin chain [9,10,38,47–54]. The fact that the profiles are functions of the scaling variable $x/t$ raises the possibility of a hydrodynamic or semiclassical description in which the space-time dependence of various quantities can be understood in terms of ballistically propagating quasiparticles [55]. Recently this description has been improved further into a kind of generalized hydrodynamics [53,56–58] suitable to describe ballistic transport in Bethe Ansatz integrable systems. The notion of emerging eigenstate solutions [59,60] constitutes another interesting direction.

In this work we present a thorough analysis of the partitioning protocol for the transverse field Ising model, a paradigmatic integrable quantum spin chain. This system can be mapped to free fermions, which makes it possible to give a microscopic derivation of the space-time profiles in inhomogeneous quenches. First principles derivations of this kind provide a valuable testing ground for more general frameworks as the hydrodynamic approach. Somewhat surprisingly, apart from the energy current and its fluctuations in the NESS [16], not much is known about the time evolution and the approach to the NESS. Only the (transverse) magnetization profiles have been computed for domain wall initial states [36,43] and for a special two-temperature initial state at the quantum critical point [40]. The work [42] went beyond the NESS and studied the space and time dependent correlation matrix of Majorana fermions in a setup slightly different form ours.

In this paper we fill this gap by providing analytical expressions for the profiles of various quantities at or away from the critical point in both the paramagnetic and ferromagnetic phase, and for a broad class of initial states defined by the initial fermionic occupations. As a byproduct, this provides a derivation of the NESS using the standard toolkit of many-body physics. Compared to the currents and other expectation values, two-point correlation functions are much less studied (notable exceptions are [15,38,39,46,50,61]), here we discuss these as well. Note that due to the non-locality of the mapping to free fermions, the interacting nature of the model shows up in correlation functions.

We also go beyond the hydrodynamic scaling limit by computing the leading corrections to the NESS corresponding to the late time approach to the steady state. These corrections are very difficult to determine in general, and the Ising model gives us the rare opportunity to derive them analytically. This can provide insights to the applicability of (generalized) hydrodynamic approaches based on the local density approximation to describe corrections to

the scaling limit.

We also study the fine structure of the propagating front which for free fermions shows universal features and a characteristic subdiffusive scaling related to the Airy kernel [39, 62, 63]. It is an interesting question how much of these features appear in the more complicated Ising model.

The paper is organized as follows. In Sec. 2 we review the diagonalization of the Ising chain with open boundary conditions. In Secs. 3-5 we analyze the non-equilibrium dynamics of the fermionic correlators, the building blocks of all spin correlation functions. In Sec. 3 we discuss the computation of the time evolution of fermionic correlation functions. The resulting integral expressions are analyzed in the semiclassical limit in Sec. 4 yielding the profiles of the fermionic correlators. In Sec. 5 we focus on the late time approach of the NESS and on the fine structure of the front. In Sec. 6 we use the results for the fermionic correlation functions obtained in Secs. 3-5 to derive the profiles of various physical quantities, including the energy density and current, the transverse magnetization and spin-spin correlation functions. The domain wall initial state is discussed in Sec. 7. We give our summary and outlook in Sec. 8. Details of the calculations and plots in support of the analytical results are collected in the appendices.

## 2 Diagonalizing the Ising spin chain

We start by giving a brief summary of the diagonalization of the transverse field Ising chain with open boundary conditions through a non-local mapping to free fermions [64,65]. Further details are given in Appendix A.

The Hamiltonian of the transverse field Ising spin chain is

$$H = -J \sum_{j=1}^{L-1} \sigma_j^x \sigma_{j+1}^x - Jh \sum_{j=1}^{L} \sigma_j^z, \tag{1}$$

where $\sigma_j^\alpha$ are the Pauli matrices, and we assume open boundary conditions. $J$ is the strength of the Ising coupling, and $h$ sets the transverse field. Energies and times are measured in $J$ and $J^{-1}$, respectively. We set $J = 1/2$ and work with dimensionless quantities throughout the paper.

The Hamiltonian (1) can be rewritten in terms of canonical spinless fermions $c_j, c_j^\dagger$ after a Jordan–Wigner transformation:

$$c_j = \prod_{k=1}^{j-1} (-\sigma_k^z) \sigma_j^- = \prod_{k=1}^{j-1} e^{i\pi \sigma_k^+ \sigma_k^-} \sigma_j^-, \qquad c_j^\dagger = \prod_{k=1}^{j-1} (-\sigma_k^z) \sigma_j^+ = \prod_{k=1}^{j-1} e^{-i\pi \sigma_k^+ \sigma_k^-} \sigma_j^+, \tag{2}$$

where $\sigma_j^\pm = (\sigma_j^x \pm i\sigma_j^y)/2$ are the usual ladder operators, yielding

$$H = -\frac{1}{2} \sum_{j=1}^{L-1} \left[ c_j^\dagger c_{j+1} + c_{j+1}^\dagger c_j + c_j^\dagger c_{j+1}^\dagger + c_{j+1} c_j \right] - h \sum_{j=1}^{L} \left( c_j^\dagger c_j - \frac{1}{2} \right). \tag{3}$$

It turns out to be useful to introduce the combinations[2]

$$A_j = c_j^\dagger + c_j, \qquad B_j = c_j^\dagger - c_j, \tag{4}$$

---

[2]In the literature it is common to include a factor of $i$ in the definition of $B_j$ so that both $A_j$ and $B_j$ are Majorana operators. Here we use the original definitions of Ref. [64].

in terms of which

$$H = \frac{1}{2} \sum_{j=1}^{L-1} A_{j+1} B_j + \frac{h}{2} \sum_{j=1}^{L} A_j B_j . \tag{5}$$

The Hamiltonian (3) can be diagonalized by a linear transformation leading to the modes $\{\eta_k, \eta_k^\dagger\}$:

$$\eta_k = \frac{1}{2} \sum_{j=1}^{L} \left[ \phi_k(j) A_j - \psi_k(j) B_j \right], \qquad A_j = \sum_k \phi_k(j)(\eta_k^\dagger + \eta_k), \tag{6a}$$

$$\eta_k^\dagger = \frac{1}{2} \sum_{j=1}^{L} \left[ \phi_k(j) A_j + \psi_k(j) B_j \right], \qquad B_j = \sum_k \psi_k(j)(\eta_k^\dagger - \eta_k). \tag{6b}$$

The (real) mode functions are

$$\phi_k(j) = A_k \sin(kj - \theta_k), \tag{7a}$$
$$\psi_k(j) = -A_k \sin(kj), \tag{7b}$$

where $A_k$ is a normalization constant such that $\sum_{j=1}^{L} \phi_k(j)^2 = \sum_{j=1}^{L} \psi_k(j)^2 = 1$, and $\theta_k$ is the Bogoliubov angle

$$\tan \theta_k = \frac{\sin k}{h + \cos k} , \tag{8}$$

where we select the branch such that $\theta_k \in [0, \pi)$. The variable $k$ is quantized as

$$k(L+1) - \theta_k = n\pi, \qquad n \in \mathbb{Z}, \tag{9}$$

implying $\phi_k(j) = -A_k(-1)^n \sin[k(L+1-j)]$.

In terms of the modes $\eta_k$ the Hamiltonian reads

$$H = \sum_k \varepsilon_k \eta_k^\dagger \eta_k + \text{const.}, \tag{10}$$

where the energy eigenvalues are[3]

$$\varepsilon_k = \sqrt{1 + 2h \cos k + h^2} . \tag{11}$$

We note that for $h < L/(L+1)$ the quantization condition (9) has a complex root corresponding to a localized edge mode which however does not affect the results below obtained in the $L \to \infty$ limit.[4] The group velocity of these excitations is

$$v_k = \frac{d\varepsilon_k}{dk} = -\frac{h \sin k}{\varepsilon_k} . \tag{12}$$

It follows that the maximal velocity is $h$ for $h < 1$ and $1$ for $h \geq 1$,

$$v_{\text{max}} = \min(1, h). \tag{13}$$

---

[3]Our expressions for $\theta_k$ and $\varepsilon_k$ differ in minus signs from many works on the subject. This comes from a difference in the mapping (2), in particular, dropping the minus signs in the Jordan–Wigner string amounts to a $\pi$-shift in the definition of the momentum $k$.

[4]The edge modes can be more important for other initial states.

## 3  Time evolution of the fermionic correlation functions

The inhomogeneous initial state consists of two independent, disjoint chains of length $L$ having macroscopically different properties, e.g. thermalized at different temperatures $T_L$ and $T_R$. So the initial density matrix is a tensor product,

$$\rho_0 = \rho_L \otimes \rho_R. \tag{14}$$

At time $t = 0$ the two chains are joined and let evolve by the Hamiltonian $H$ of the chain of length $2L$. In other words, we turn on the coupling between site 0 and site 1,

$$H = H_L + H_R - \frac{1}{2}\sigma_0^x \sigma_1^x = \sum_k \varepsilon_k \gamma_k^\dagger \gamma_k + \text{const.}, \tag{15}$$

where $H_L$ is the Hamiltonian of the left chain of sites $j = -L+1, \ldots, 0$, $H_R$ is the Hamiltonian of the right chain of sites $j = 1, \ldots, L$. This setup is sometimes referred to as the "two-reservoir initial state" or the "partitioning protocol". In Eq. (15) we introduced the fermionic mode operators $\gamma_k$ that diagonalize $H$ on the chain of length $2L$.

Let us denote the mode functions of $H_R$ by $\phi_q^R$ and $\psi_q^R$, then due to swapping the boundary conditions, the mode functions of $H_L$ are given by

$$\phi_q^L(j) = \psi_q^R(1-j), \qquad \psi_q^L(j) = \phi_q^R(1-j). \tag{16}$$

The mode functions of the final Hamiltonian on the full chain $H$ are $\varphi_k(j) = \phi_k(j+L)$ and $\chi_k(j) = \psi_k(j+L)$, where the momenta $\{k_n\}$ are quantized in volume $2L$, which amounts to the substitution $L \to 2L$ in Eq. (9). Note that the functional form of $\varepsilon_k$ and $\theta_k$ are unchanged as we do not perform a quench in the Ising interaction or the transverse field. The fermion operators $A_j, B_j$ on the full chain are related to the modes by

$$A_j = \sum_k \varphi_k(j)(\gamma_k^\dagger + \gamma_k), \qquad B_j = \sum_k \chi_k(j)(\gamma_k^\dagger - \gamma_k). \tag{17}$$

In order to compute the time evolution of spin correlation functions, we first need to compute the building blocks given by the correlations $\langle A_n(t) A_m(t) \rangle$, $\langle A_n(t) B_m(t) \rangle$, $\langle B_n(t) B_m(t) \rangle$. The time evolved operators in the Heisenberg picture are obtained by writing them in terms of $\gamma_k$ using (17), exploiting the simple time dependence of the mode operators and then rewriting everything in terms of $A_j, B_j$. After these steps we find [36]

$$A_n(t) = \sum_j \langle A_j | A_n(t) \rangle A_j + \langle B_j | A_n(t) \rangle B_j, \tag{18a}$$

$$B_n(t) = \sum_j \langle A_j | B_n(t) \rangle A_j + \langle B_j | B_n(t) \rangle B_j \tag{18b}$$

with coefficients

$$\langle A_j | A_n(t) \rangle = \sum_k \varphi_k(j) \varphi_k(n) \cos(\varepsilon_k t), \tag{19a}$$

$$\langle B_j | B_n(t) \rangle = \sum_k \chi_k(j) \chi_k(n) \cos(\varepsilon_k t), \tag{19b}$$

$$\langle A_j | B_n(t) \rangle = \langle B_n | A_j(t) \rangle = i \sum_k \varphi_k(j) \chi_k(n) \sin(\varepsilon_k t). \tag{19c}$$

In the infinite volume limit, $L \to \infty$, the sum over $k$ can be written as an integral. After dropping highly oscillating terms in the integrands[5] we find

$$\langle A_j | A_n(t) \rangle = \langle B_j | B_n(t) \rangle = \int_{-\pi}^{\pi} \frac{dk}{2\pi} \tilde{\varphi}_k^*(j) \tilde{\varphi}_k(n) \cos(\varepsilon_k t), \tag{20a}$$

$$\langle A_j | B_n(t) \rangle = \langle B_n | A_j(t) \rangle = i \int_{-\pi}^{\pi} \frac{dk}{2\pi} \tilde{\varphi}_k^*(j) \tilde{\chi}_k(n) \sin(\varepsilon_k t), \tag{20b}$$

where we defined the infinite volume mode functions

$$\tilde{\varphi}_k(j) = e^{-ikj + i\theta_k}, \qquad \tilde{\chi}_k(j) = -e^{-ikj}. \tag{21}$$

Using Eqs. (18), the time-evolved two-point functions can be written as

$$\langle X_n(t) Y_m(t) \rangle = \sum_{j,l} \Big[ \langle A_j | X_n(t) \rangle \langle A_l | Y_m(t) \rangle \langle A_j A_l \rangle_0 + \langle A_j | X_n(t) \rangle \langle B_l | Y_m(t) \rangle \langle A_j B_l \rangle_0 \\ + \langle B_j | X_n(t) \rangle \langle A_l | Y_m(t) \rangle \langle B_j A_l \rangle_0 + \langle B_j | X_n(t) \rangle \langle B_l | Y_m(t) \rangle \langle B_j B_l \rangle_0 \Big], \tag{22}$$

where $X$ and $Y$ stand for $A$ or $B$. The initial correlations will be non-zero only if $j, l \geq 1$ or $j, l \leq 0$, so the correlation function (22) can be split into two parts corresponding to the contributions of the left and right chains. We shall denote these contributions by $\langle X_n(t) Y_m(t) \rangle^{\mathrm{L/R}}$. The initial correlations are calculated by rewriting the fermion operators in terms of the mode operators $\eta_q$ that diagonalize the left and right half chains. Here we assume that for each half chain the anomalous correlations are zero, so

$$\langle \eta_q^\dagger \eta_{q'} \rangle_0 = \delta_{q,q'} f_q, \tag{23a}$$

$$\langle \eta_q^\dagger \eta_{q'}^\dagger \rangle_0 = \langle \eta_q \eta_{q'} \rangle_0 = 0. \tag{23b}$$

For the two-temperature setup

$$f_q = \frac{1}{1 + e^{\varepsilon_q / T_{\mathrm{L/R}}}} \tag{24}$$

is the thermal Fermi–Dirac distribution function. Exploiting the completeness of the mode functions, we arrive at

$$\langle A_j A_l \rangle_0 = -\langle B_j B_l \rangle_0 = \delta_{j,l}, \tag{25a}$$

$$\langle A_j B_l \rangle_0 = -\langle B_l A_j \rangle_0 = \sum_q \phi_q^{\mathrm{R}}(j) \psi_q^{\mathrm{R}}(l)(1 - 2f_q^{\mathrm{R}}) \qquad j, l \geq 1, \tag{25b}$$

$$\langle A_j B_l \rangle_0 = -\langle B_l A_j \rangle_0 = \sum_q \phi_q^{\mathrm{L}}(j) \psi_q^{\mathrm{L}}(l)(1 - 2f_q^{\mathrm{L}}) \qquad j, l \leq 0. \tag{25c}$$

Due to the orthonormality of the mode functions, the terms containing the correlations (25a) yield a Kronecker $\delta_{n,m}$ in Eq. (22), resulting in

$$\langle X_n(t) Y_m(t) \rangle = \Delta_{n,m} + \sum_{j,l} \Big[ \langle A_j | X_n(t) \rangle \langle B_l | Y_m(t) \rangle - \langle A_j | Y_m(t) \rangle \langle B_l | X_n(t) \rangle \Big] \langle A_j B_l \rangle_0, \tag{26}$$

where

$$\Delta_{n,m} = \begin{cases} \delta_{n,m} & X = Y = A, \\ -\delta_{n,m} & X = Y = B, \\ 0 & \text{otherwise.} \end{cases} \tag{27}$$

---

[5] These are terms in which $kL$ appears in the argument of the trigonometric functions which due to Eq. (9) yields alternating signs for neighboring $k$ values.

Since the two half-chains initially differ solely in the initial fermionic occupations, the left/right parts of the fermionic correlation functions are related to each other by (for a proof, see Appendix B)

$$\langle A_n(t)A_m(t)\rangle = \langle B_{1-m}(t)B_{1-n}(t)\rangle|_{f^{\mathrm{L}}\leftrightarrow f^{\mathrm{R}}}, \tag{28a}$$

$$\langle A_n(t)B_m(t)\rangle = \langle A_{1-m}(t)B_{1-n}(t)\rangle|_{f^{\mathrm{L}}\leftrightarrow f^{\mathrm{R}}}. \tag{28b}$$

Expression (26) can be used to numerically evaluate all equal-time correlation functions on a finite chain by using the finite volume mode functions (7) as well as the edge modes in the initial correlations (25) and in the coefficients (19). All our numerical results presented in the forthcoming sections below were obtained in this way.

Equation (26) is also the starting point of analytic manipulations in the infinite system size limit. Then the coefficients in Eqs. (20) are used and the sums in Eqs. (25) are straightforwardly turned into integrals. The remaining sums over lattice sites run from $-\infty$ to $\infty$, eliminating all explicit dependence on $L$. The correlation functions are thus given by a double sum of triple integrals. The sums are essentially geometric series and can be performed analytically. The integral over $q$ coming from expressions (25) can be manipulated using contour integral techniques. For details of the derivation we refer the reader to Appendix B. The final result is a set of double integral expression that give all the equal-time correlations *exactly* at any time after the quench:

$$\langle A_n(t)B_m(t)\rangle = -\langle B_m(t)A_m(t)\rangle$$
$$= -\frac{i}{2}\int_{-\pi}^{\pi}\frac{\mathrm{d}k}{2\pi}\int_{-\pi}^{\pi}\frac{\mathrm{d}k'}{2\pi}G^{\mathrm{R}}(k,k')e^{i\theta_k}\frac{\varepsilon_k\cos(\varepsilon_k t)\cos(\varepsilon_{k'}t)+\varepsilon_{k'}\sin(\varepsilon_k t)\sin(\varepsilon_{k'}t)}{\cos(k'-i\delta)-\cos(k+i\delta)}e^{i(k'm-kn)}$$
$$-\frac{i}{2}\int_{-\pi}^{\pi}\frac{\mathrm{d}k}{2\pi}\int_{-\pi}^{\pi}\frac{\mathrm{d}k'}{2\pi}G^{\mathrm{L}}(k,k')e^{i\theta_k}\frac{\varepsilon_k\cos(\varepsilon_k t)\cos(\varepsilon_{k'}t)+\varepsilon_{k'}\sin(\varepsilon_k t)\sin(\varepsilon_{k'}t)}{\cos(k'-i\delta)-\cos(k+i\delta)}e^{i(km-k'n)}e^{i(k'-k)},$$

$$\tag{29}$$

$$\langle A_n(t)A_m(t)\rangle - \delta_{n,m}$$
$$= -\frac{1}{2}\int_{-\pi}^{\pi}\frac{\mathrm{d}k}{2\pi}\int_{-\pi}^{\pi}\frac{\mathrm{d}k'}{2\pi}G^{\mathrm{R}}(k,k')e^{i(\theta_k-\theta_{k'})}\frac{\varepsilon_k\cos(\varepsilon_k t)\sin(\varepsilon_{k'}t)-\varepsilon_{k'}\sin(\varepsilon_k t)\cos(\varepsilon_{k'}t)}{\cos(k'-i\delta)-\cos(k+i\delta)}e^{i(k'm-kn)}$$
$$-\frac{1}{2}\int_{-\pi}^{\pi}\frac{\mathrm{d}k}{2\pi}\int_{-\pi}^{\pi}\frac{\mathrm{d}k'}{2\pi}G^{\mathrm{L}}(k,k')e^{i(k'-k)}\frac{\varepsilon_k\sin(\varepsilon_k t)\cos(\varepsilon_{k'}t)-\varepsilon_{k'}\cos(\varepsilon_k t)\sin(\varepsilon_{k'}t)}{\cos(k'-i\delta)-\cos(k+i\delta)}e^{i(km-k'n)}.$$

$$\tag{30}$$

Here ($\nu = $ L,R)

$$G^{\nu}(k,k') = g^{\nu}(k) - g^{\nu}(-k'), \qquad g^{\nu}(k) = \frac{\sin k}{\varepsilon_k}(1-2f_k^{\nu})+I^{\nu}(k), \tag{31}$$

where $I^{\nu}(k)$ are even functions, expressed as contour integrals in Eq. (158) of Appendix B. The $\langle B_n B_m\rangle$ correlations can be obtained using relations (28).

These expressions were not known and thus are new results of the paper. They can be used to obtain further analytic results in various limits which is the subject of the next section.

## 4 The semiclassical limit

The existence of a scaling limit for the magnetization profile evolving from a domain wall initial state in the XX spin chain was first observed in Ref. [34]. The same behavior was found for the two-temperature initial state [12] and for the Ising chain [36, 42].

This semiclassical or hydrodynamic limit corresponds to the case when $n, m, t \to \infty$ with $\lim n/t = \lim m/t$ fixed implying $(n-m)/t \to 0$. Physically, we focus on the large time behavior near a given *ray* defined by the ratio $n/t \approx m/t$. The appearance of this scaling variable suggests a hydrodynamic description in terms of quasiparticles [55].

The behavior in this limit can be derived through a stationary phase analysis of the double integral expressions [12,39]. In each integral, the stationary points are given by

$$v_{k_s} = \frac{d\varepsilon(k_s)}{dk_s} = \pm n/t \,, \qquad\qquad v_{k'_s} = \pm m/t \,. \tag{32}$$

As $(n-m)/t \to 0$, the stationary points coalesce, $k_s \to k'_s$, and the integrand becomes singular. This singularity governs the leading order behavior in the semiclassical limit. We note that another singularity could arise as $k_s \to -k'_s$, however, the integrand vanishes due to $G^\nu(k, -k) = 0$ so this gives a subleading contribution.

Let us start with the right contribution to the correlation (29) given by the first line of the right hand side. We first introduce the center of mass and relative coordinates along with the momentum difference and total momentum:

$$x = (n+m)/2, \qquad\qquad K = (k+k')/2, \tag{33a}$$
$$r = m-n, \qquad\qquad Q = k-k'. \tag{33b}$$

Now we expand around $Q = 0$ to find

$$\langle A_n(t)B_m(t)\rangle^{\mathrm{R}} \approx \frac{1}{2}\int_{-\pi}^{\pi}\frac{dK}{2\pi}\int_{-2\pi}^{2\pi}\frac{dQ}{2\pi i}\frac{2\sin K}{\varepsilon_K}(1-2f_K^{\mathrm{R}})e^{i(\theta_K+\theta'_K Q/2)}e^{i(Kr-Qx)}\frac{\cos(\varepsilon'_K Qt)\varepsilon_K}{\sin K(Q+2i\delta)} \,. \tag{34}$$

Using the integral expression for the Heaviside theta function, $\int_{-\infty}^{\infty}\frac{dQ}{2\pi i}\frac{e^{izQ}}{Q+i\delta} = -\Theta(-z)$, we obtain

$$\langle A_n(t)B_m(t)\rangle_{\mathrm{sc}}^{\mathrm{R}} = \int_{-\pi}^{\pi}\frac{dK}{2\pi}(1-2f_K^{\mathrm{R}})e^{i\theta_K}e^{iKr}\frac{1}{2}[-\Theta(x-\varepsilon'_K t-\theta'_K/2)-\Theta(x+\varepsilon'_K t-\theta'_K/2)]. \tag{35}$$

In the semiclassical limit $x, t \to \infty$, so the constant shift in the argument of the step functions can be neglected. The left contribution can be obtained from Eqs. (28) via the substitution $n \to 1-m, m \to 1-n$ which implies $x \to 1-x \approx -x, r \to r$. The total result is then

$$\langle A_n(t)B_m(t)\rangle_{\mathrm{sc}} = -\int_{-\pi}^{\pi}\frac{dK}{2\pi}\cos(Kr+\theta_K)[(1-2f_K^{\mathrm{R}})\Theta(u-v_K)+(1-2f_K^{\mathrm{L}})\Theta(-u+v_K)], \tag{36}$$

where $u = x/t = (n+m)/(2t)$ specifies the ray. In a similar fashion, from (30) we obtain

$$\langle A_n(t)A_m(t)\rangle_{\mathrm{sc}} = -\langle B_n(t)B_m(t)\rangle_{\mathrm{sc}} = \delta_{n,m} + 2i\int_{-\pi}^{\pi}\frac{dK}{2\pi}\sin(Kr)[f_K^{\mathrm{R}}\Theta(u-v_K)+f_K^{\mathrm{L}}\Theta(-u+v_K)], \tag{37}$$

where we used that the integral of $\sin(Kr)$ vanishes.

For a fixed separation $r$, the resulting expressions (36),(37) are functions of $u = x/t$, i.e. they depend on the ray only, which is the hallmark of ballistic behavior. Indeed, a natural interpretation of these expressions can be given in terms of free quasiparticles [55] with dispersion relation $\varepsilon_k$ that are present on each side before the quench following the original left and right distributions. Around a given space-time point $(x, t)$ only those left particles can contribute which have velocities larger than $u = x/t$. Similarly, contributing right particles cannot have velocities greater than $u$. This is perfectly reflected by the Heaviside theta functions of the integrands. The same observation was made in [42] for translationally invariant Gaussian initial

states evolving under a Hamiltonian with a localized defect. As we will see below, this interpretation is even clearer for simple physical quantities such as the energy density and energy current.

If $u > v_{\text{max}} = \min(1, h)$, or $u < -v_{\text{max}}$, then one of the theta functions in the integrands is equal to 1 while the other vanishes for all $K$, so only the right or the left contribution survives. This is a horizon effect: the physics outside of a light cone set by the maximal group velocity is unaffected by the quench of joining the two sides. If the initial half chains were in equilibrium before the quench then all observables assume their initial expectation values and they change only when the ballistically propagating front arrives.[6]

For asymptotically large times, a non-equilibrium steady state (NESS) is formed around the junction in the middle of the chain. In accordance with the quasiparticle picture, the region in which the system is asymptotically close to the NESS grows ballisitically. The NESS is obtained in the limit $t \to \infty$ and $n, m$ fixed which amounts to $u = 0$, implying

$$\langle A_n B_m \rangle_{\text{NESS}} = -\int_0^{\pi} \frac{dK}{2\pi} \cos(Kr + \theta_K)(1 - 2f_K^{\text{R}}) - \int_{-\pi}^0 \frac{dK}{2\pi} \cos(Kr + \theta_K)(1 - 2f_K^{\text{L}})$$
$$= -\int_{-\pi}^{\pi} \frac{dK}{2\pi} e^{i(Kr + \theta_K)} \left(1 - 2\frac{f_K^{\text{R}} + f_K^{\text{L}}}{2}\right), \quad (38)$$

and

$$\langle A_n A_m \rangle_{\text{NESS}} - \delta_{n,m} = -\langle B_n B_m \rangle_{\text{NESS}} - \delta_{n,m}$$
$$= 2i \int_0^{\pi} \frac{dK}{2\pi} \sin(Kr) f_K^{\text{R}} + 2i \int_{-\pi}^0 \frac{dK}{2\pi} \sin(Kr) f_K^{\text{L}} = \int_{-\pi}^{\pi} \frac{dK}{2\pi} e^{iKr} \left(f_K^{\text{R}} - f_K^{\text{L}}\right) \text{sgn}(K). \quad (39)$$

As the NESS is translationally invariant, the correlations in the NESS depend only on the separation $r = m - n$. The NESS correlations can be interpreted in terms of quasiparticles with the left momentum distribution (e.g. thermalized with the left temperature) going to the right and by quasiparticles with the right momentum distribution (e.g. thermalized with the right temperature) going to the left.

These expressions were first derived using $C^*$-algebra methods in [14,66]. In our approach, the NESS appears as a single special member of a continuous family that gives the space-time profile of correlations in the semiclassical limit. The same was achieved in [42] for a homogeneous initial state evolving under a non-homogeneous Hamiltonian.

Finally, let us write down the steady state correlations of the original fermion operators:

$$\langle c_l c_m \rangle_{\text{NESS}} = -\int_0^{\pi} \frac{dk}{2\pi} (1 - f_k^{\text{R}} - f_k^{\text{L}}) \sin \theta_k \sin[k(m - l)], \quad (40)$$

$$\langle c_l^{\dagger} c_m \rangle_{\text{NESS}} = \frac{\delta_{l,m}}{2} + \int_0^{\pi} \frac{dk}{2\pi} (1 - f_k^{\text{R}} - f_k^{\text{L}}) \cos \theta_k \cos[k(m - l)]$$
$$+ i \int_0^{\pi} \frac{dk}{2\pi} (f_k^{\text{R}} - f_k^{\text{L}}) \sin[k(m - l)]. \quad (41)$$

## 5 Beyond the semiclassical limit

In this section we discuss the leading finite time corrections to the NESS and the structure of the propagating front.

---

[6]If the initial states of the half chains are not equilibrium states of the left/right Hamiltonians, then the dynamics are non-trivial outside the light cone, but unaffected by the other side. For a simple example see Sec. 7.

## 5.1 Approach to the NESS

Although the correlations in the NESS were known, the finite time corrections characterizing the late time approach to the NESS have not been analyzed before. The analytic expressions (29), (30) can be used to study the leading corrections in the limit $t \to \infty$, $n/t, m/t \to 0$. The details of the calculations are given in Appendix D, here we only quote the results.

For $\langle A_n B_m \rangle^{\mathrm{R}}$ we find

$$
\begin{aligned}
\langle A_n(t) B_m(t) \rangle^{\mathrm{R}} = {} & \langle A_n B_m \rangle^{\mathrm{R}}_{\mathrm{NESS}} \\
& + i \frac{\sqrt{|1-h^2|}}{2\pi t} [g^{\mathrm{R}}(0) - g^{\mathrm{R}}(\pi)] \left[ \frac{(-1)^m + \mathrm{sgn}(h-1)(-1)^n}{2} \frac{\sin[2\min(1,h)t]}{2\min(1,h)} \right. \\
& \left. \hphantom{+ i \frac{\sqrt{|1-h^2|}}{2\pi t}} + \frac{(-1)^m - \mathrm{sgn}(h-1)(-1)^n}{2} \frac{\cos[2\max(1,h)t]}{2\max(1,h)} \right] \\
& - \frac{1}{4\pi h t} \Big[ (1+h)^2 [g^{\mathrm{R}\prime}(0)(m+n-\theta_0') + i g^{\mathrm{R}\prime\prime}(0)] \\
& \hphantom{- \frac{1}{4\pi h t}} - \mathrm{sgn}(h-1)(-1)^{m-n}(1-h)^2 [g^{\mathrm{R}\prime}(\pi)(m+n-\theta_\pi') + i g^{\mathrm{R}\prime\prime}(\pi)] \Big] + \mathcal{O}\left(\frac{1}{t^2}\right). \quad (42)
\end{aligned}
$$

Note that in the critical case ($h = 1$) the result is much simpler,

$$
\langle A_n(t) B_m(t) \rangle^{\mathrm{R}} = \langle A_n B_m \rangle^{\mathrm{R}}_{\mathrm{NESS}} - \frac{1}{\pi t} [g^{\mathrm{R}\prime}(0)(m+n-1/2) + i g^{\mathrm{R}\prime\prime}(0)], \quad (43)
$$

where we used that $\theta_0' = 1/2$. The correction to the left part $\langle A_n B_m \rangle^{\mathrm{L}}_{\mathrm{NESS}}$ can be obtained from Eq. (42) via the mapping (28). These expressions are checked by comparing them to numerical results in Fig. 7 in Appendix D.

The first correction in the second and third line is oscillating around the NESS value both in time and space. The wavelength of the spatial oscillation is 2 (lattice spacings) while the frequency of the temporal oscillation is either 2 or $2h$. Some of the terms in the last two lines can be written as

$$
-\frac{m+n}{4\pi h t} \Big[ (1+h)(1-2f_0) - (-1)^{m-n}(1-h)(1-2f_\pi) \Big], \quad (44)
$$

where for the derivatives of $g(k)$ and $\theta_k$ we used Eqs. (174),(187) and that $\theta(\pi) = 0$ for $h > 1$ and $\theta(\pi) = \pi$ for $h < 1$. This means that the profile of $\langle A_n B_m \rangle$ for fixed time and separation $m-n$ is linear near the origin. Since $(m+n)/t = 2u$, this is the first order correction in $u$ to the NESS given by $u = 0$. Indeed, (44) can be obtained by a Taylor expansion of the semiclassical result (36) in $u$ (see Appendix D).

On top of this there is an additional correction in the last two lines of (42):

$$
-\frac{1}{4\pi h t} \Big[ -1 + 2f_0 + i(1+h)^2 g^{\mathrm{R}\prime\prime}(0) + (-1)^{m-n}(-1 + 2f_\pi + i(1-h)^2 g^{\mathrm{R}\prime\prime}(\pi)) \Big]. \quad (45)
$$

This amounts to a finite shift at any large but finite $t$, in particular, in the origin the NESS is not reached immediately but only asymptotically as $\sim 1/t$.

All these different types of corrections can be observed in Figs. 1b,1d where we plotted the energy density, a combination of $\langle A_n B_m \rangle$ correlators (c.f. Eq. (63)). With time, the flat profile of the translationally invariant NESS develops: the oscillations get damped, and both the slope and the offset of the profile near the origin tends to zero.

In a similar fashion, we obtain

$$
\begin{aligned}
\langle A_n(t) A_m(t) \rangle^{\mathrm{R}} = {} & \langle A_n A_m \rangle^{\mathrm{R}}_{\mathrm{NESS}} + [(-1)^m - (-1)^n] \frac{\sqrt{|1-h^2|}}{4\pi t} [g^{\mathrm{R}}(0) - g^{\mathrm{R}}(\pi)] \mathrm{sgn}(h-1) \\
& \times \left( \frac{\sin[2\max(1,h)t]}{2\max(1,h)} + \frac{\cos[2\min(1,h)t]}{2\min(1,h)} \right) + \mathcal{O}(t^{-2}), \quad (46a)
\end{aligned}
$$

and

$$\langle A_n(t)A_m(t)\rangle^{\mathrm{L}} = \langle A_n A_m\rangle^{\mathrm{L}}_{\mathrm{NESS}} + \left[(-1)^m - (-1)^n\right]\frac{\sqrt{|1-h^2|}}{4\pi}\left[g^{\mathrm{L}}(0) - g^{\mathrm{L}}(\pi)\right]$$
$$\times \left(\frac{\sin[2\max(1,h)t]}{2\max(1,h)t} - \frac{\cos[2\min(1,h)t]}{2\min(1,h)t}\right) + \mathscr{O}(t^{-2}). \quad (46b)$$

The corresponding expressions for $\langle B_n(t)B_m(t)\rangle$ can be determined using the mapping (28). These corrections are compared with the numerical data in Fig. 7. In contrast to $\langle A_n B_m\rangle$, the correlation functions $\langle A_n A_m\rangle$ and $\langle B_n B_m\rangle$ are constant around the origin and are not shifted with respect to their NESS value. The leading order corrections are given by oscillations whose amplitude decays as $\sim 1/t$. However, note that if $n-m$ is even, these terms vanish and the leading correction is of order $\mathscr{O}(t^{-2})$. This is also true in the critical case $h=1$ for any $n,m$.

We note that the leading order corrections in Eqs. (42),(46) show a parity effect for $h \neq 1$, i.e. oscillations with wavelength of two lattice sites. This may cast doubt on the ability of the local density approximation and a hydrodynamic description to capture the corrections to the ballistic semiclassical results.[7]

Finally, we mention that a very similar derivation can be performed in principle at finite $u = x/t$ values to obtain the leading late time corrections to the semiclassical profile. The essential difference is that the stationary points would be shifted. We have not carried out this calculation but performed numerical checks which show that the leading corrections are $O(1/t)$ also in this case.

## 5.2 Fine structure of the front

It is an interesting question whether the fine structure of the propagating front shows some universal behavior. In Ref. [62] the XX spin chain evolving from a domain wall initial state was studied, and it was found that the shape of the front can be described by a scaling function. In particular, the width of the front scales sub-diffusively, as $t^{1/3}$. In Ref. [63] the full distribution of the magnetization was determined and it was related to the eigenvalue statistics of random matrices. The rescaled correlations were shown to be described by the Airy kernel (see also [39]). In Ref. [40] a similar behavior was observed numerically for the two-temperature setup in the Ising spin chain for $h > 1$ and $T_{\mathrm{L}} = 0, T_{\mathrm{R}} = \infty$.

Here we analyze this edge behavior within our formalism for any $h$ and any left/right temperatures, following the method of Ref. [39]. The fine structure of the front is given by a correction to the time-independent equilibrium value in the region that has not yet been reached by the front. This correction can be studied by taking the limit $x, t \to \infty$, $x/t, y/t \approx v_{\mathrm{max}}$, $(x-y)/t \to 0$. In this limit the stationary points for $x$ and $y$ approach each other giving rise to a specific scaling form of the edge of the front.

We present the analysis of $\langle A_n(t)B_m(t)\rangle^{\mathrm{R}}$ near the right front. This is the most complicated case, as the right equilibrium value of this correlation is non-zero (The left contributions are zero outside of the light cone and the $\langle A_n A_m\rangle$, $\langle B_n B_m\rangle$ correlations are zero for $n \neq m$.) Clearly, this value must be subtracted before we can focus on the correction. The starting point is the first double integral expression in the general formula (29). We show that the (right) equilibrium correlation is related to the residue of the integrand at the pole $k = k'$. Indeed, using $G^{\mathrm{R}}(k,k) = 2\sin k/\varepsilon_k(1-2f_k^{\mathrm{R}})$ we get

$$\frac{1}{2}\int_{-\pi}^{\pi}\frac{\mathrm{d}k}{2\pi}\frac{2\sin k}{\varepsilon_k}(1-2f_k^{\mathrm{R}})e^{i\theta_k}\varepsilon_k\frac{\cos^2(\varepsilon_k t) + \sin^2(\varepsilon_k t)}{\sin k}e^{ik(m-n)} = \int_{-\pi}^{\pi}\frac{\mathrm{d}k}{2\pi}(1-2f_k^{\mathrm{R}})e^{i\theta_k}e^{ik(m-n)},$$
$$(47)$$

---

[7]We thank one of the referees for pointing out this aspect.

which coincides with the semiclassical result for $u = (n+m)/(2t) > v_{\max}$ that gives the equilibrium expectation value to the right of the front.

In the double integrals in (29) the infinitesimal imaginary shifts in the denominator mean that the integration contour of the $k'$ integral avoids the pole at $k' = k$ from *below*. By virtue of the residue theorem, the expression (47) is the result of performing the $k'$ integral along the contour encircling the point $k$ on the real axis. Subtracting this value is equivalent to pulling the contour through the pole at $k' = k$ so that it runs *above* the pole. This amounts to changing the sign of $\delta$ in the double integral.

Then the correction $\Delta \langle A_n(t)B_m(t)\rangle^{\mathrm{R}} \equiv \langle A_n(t)B_m(t)\rangle^{\mathrm{R}} - \langle A_n B_m\rangle^{\mathrm{R}}_{\mathrm{equil.}}$ can be recast as (note the sign of $\delta$)

$$\Delta \langle A_n(t)B_m(t)\rangle^{\mathrm{R}} = -\frac{i}{2}\int_{-\pi}^{\pi}\frac{\mathrm{d}k}{2\pi}\int_{-\pi}^{\pi}\frac{\mathrm{d}q}{2\pi}\frac{G^{\mathrm{R}}(k,q)e^{i\theta_k}}{\cos(q+i\delta)-\cos(k-i\delta)}$$
$$\left[\frac{\varepsilon_k - \varepsilon_q}{4}\left(e^{i(\varepsilon_k t - kn)}e^{i(\varepsilon_q t + qm)} + e^{-i(\varepsilon_k t + kn)}e^{-i(\varepsilon_q t - qm)}\right)\right.$$
$$\left. + \frac{\varepsilon_k + \varepsilon_q}{4}\left(e^{i(\varepsilon_k t - kn)}e^{-i(\varepsilon_q t - qm)} + e^{-i(\varepsilon_k t + kn)}e^{i(\varepsilon_q t + qm)}\right)\right]. \tag{48}$$

The stationary phase for the term in the first bracket is given by $\varepsilon'_k = n/t, \varepsilon'_q = -m/t$ or $\varepsilon'_k = -n/t, \varepsilon'_q = m/t$. As we are interested in the front, $n$ and $m$ are close to each other so $k \approx -q$ follows. But then both $\varepsilon_k - \varepsilon_q \approx 0$ and $G(k,q) \approx 0$, so this gives a subleading correction. The leading term is given by the second bracket and the stationary points $\varepsilon'_k = n/t, \varepsilon'_q = m/t$, and $\varepsilon'_k = -n/t, \varepsilon'_q = -m/t$.

Let us introduce the wave number $\kappa$ corresponding to the maximal group velocity,

$$\varepsilon'_\kappa = v_{\max} = \min(1,h) \qquad \Longrightarrow \qquad \kappa = -\arccos\left[-\min\left(h, \frac{1}{h}\right)\right]. \tag{49}$$

Then expanding around $\pm\kappa$ we have

$$\varepsilon_k t \mp kn = \varepsilon_\kappa t - \kappa n \pm \left(\varepsilon'_\kappa(k \mp \kappa) + \frac{1}{6}\varepsilon'''_\kappa(k \mp \kappa)^3 + \dots\right)t \mp (k \mp \kappa)n, \tag{50}$$

where we used that $\varepsilon''_\kappa = 0$ and that $\varepsilon_k$ is even in $k$, and analogous expressions hold for $\varepsilon_q t \pm qm$. We have to treat differently the critical and off-critical cases.

### 5.2.1 Off-critical case

For $h \neq 1$ the leading contribution can be written as

$$\Delta \langle A_n(t)B_m(t)\rangle^{\mathrm{R}} \sim -\frac{i}{2}\int\frac{\mathrm{d}\tilde{k}}{2\pi}\int\frac{\mathrm{d}\tilde{q}}{2\pi}\frac{\varepsilon_\kappa}{2}\frac{1}{\sin\kappa}$$
$$\left(G^{\mathrm{R}}(\kappa,\kappa)e^{i\theta_\kappa}e^{i\kappa(m-n)}\frac{e^{-i\tilde{k}(n-v_\kappa t)+i\tilde{q}(m-v_\kappa t)}e^{i/6\varepsilon'''_\kappa(\tilde{k}^3-\tilde{q}^3)t}}{\tilde{k}-\tilde{q}-i\delta[2+(\tilde{k}+\tilde{q})\cot\kappa]}\right.$$
$$\left. + G^{\mathrm{R}}(-\kappa,-\kappa)e^{-i\theta_\kappa}e^{-i\kappa(m-n)}\frac{e^{-i\tilde{k}(n-v_\kappa t)+i\tilde{q}(m-v_\kappa t)}e^{i/6\varepsilon'''_\kappa(\tilde{k}^3-\tilde{q}^3)t}}{-(\tilde{k}-\tilde{q}-i\delta[2-(\tilde{k}+\tilde{q})\cot\kappa])}\right), \tag{51}$$

where we defined $\tilde{k} = k - \kappa$, $\tilde{q} = q - \kappa$ in the first term and $\tilde{k} = k + \kappa$, $\tilde{q} = q + \kappa$ in the second term. As only the small $\tilde{k}, \tilde{q}$ region gives non-negligible contribution to the integral,

the $(\tilde{k} + \tilde{q})\cot\kappa$ term can be dropped in the coefficient of $i\delta$. We introduce rescaled variables,

$$X = (n - v_\kappa t)\left(\frac{-2}{\varepsilon_\kappa''' t}\right)^{1/3}, \qquad\qquad K = \left(\frac{-2}{\varepsilon_\kappa''' t}\right)^{-1/3}\tilde{k}, \qquad (52)$$

$$Y = (m - v_\kappa t)\left(\frac{-2}{\varepsilon_\kappa''' t}\right)^{1/3}, \qquad\qquad Q = \left(\frac{-2}{\varepsilon_\kappa''' t}\right)^{-1/3}\tilde{q}, \qquad (53)$$

and we extend the domain of integration to the real line since the large $K, Q$ region does not contribute due to the fast oscillations. Using $G^{\mathrm{R}}(\kappa, \kappa) = 2\sin\kappa/\varepsilon_\kappa(1 - 2f_\kappa^{\mathrm{R}})$, $\varepsilon_\kappa''' = -\min(1, h) = -v_{\max}$, we can write

$$\Delta\langle A_n(t)B_m(t)\rangle^{\mathrm{R}}$$

$$\sim \cos[\kappa(m - n) + \theta_\kappa](1 - 2f_\kappa^{\mathrm{R}})\left(\frac{2}{v_{\max}t}\right)^{1/3}\int_{-\infty}^{\infty}\frac{\mathrm{d}K}{2\pi}\int_{-\infty}^{\infty}\frac{\mathrm{d}Q}{2\pi}\frac{e^{iQY + iQ^3/3}e^{-iKX - iK^3/3}}{i(K - Q - i\delta)} \qquad (54)$$

$$= \cos[\kappa(m - n) + \theta_\kappa](1 - 2f_\kappa^{\mathrm{R}})\left(\frac{2}{v_{\max}t}\right)^{1/3}K(X, Y),$$

where

$$K(X, Y) = \frac{\mathrm{Ai}'(Y)\mathrm{Ai}(X) - \mathrm{Ai}'(X)\mathrm{Ai}(Y)}{X - Y} = \int_0^{\infty}\mathrm{d}\lambda\,\mathrm{Ai}(x + \lambda)\mathrm{Ai}(y + \lambda) \qquad (55)$$

is the celebrated Airy kernel [67].

Following the same steps we find

$$\langle A_n(t)A_m(t)\rangle^{\mathrm{R}} \sim -\langle B_n(t)B_m(t)\rangle^{\mathrm{R}} \sim \delta_{nm} - i\sin[\kappa(n - m)](1 - 2f_\kappa^{\mathrm{R}})\left(\frac{2}{v_{\max}t}\right)^{1/3}K(X, Y). \quad (56)$$

The functional form of the left contributions for all correlation functions turns out to be simply opposite in sign (and obviously, $f_\kappa^{\mathrm{R}}$ is replaced by $f_\kappa^{\mathrm{L}}$).

The appearance of the Airy kernel is quite generic in free fermionic systems near an edge originating either from a moving front as in our case, or from an external confinement [68–70]. The mathematical reason is the presence of a degenerate stationary point of the phase in the integrand of the fermionic propagator where the denominator also becomes singular [71]. Even though the integrands in question are slightly more complicated for the Ising chain, the necessary ingredients for the appearance of the Airy kernel are present. The finite temperature only enters in the prefactor of the Airy kernel.

However, not all correlations are described by the Airy kernel. Interestingly, in the ferromagnetic case ($h < 1$), $\cos\theta_\kappa = 0$ as can be seen from Eqs. (49) and (8). This implies that the right hand side of (54) vanishes for $n = m$ and the edge behavior of $\langle A_n B_n\rangle$ is *not* described by the Airy kernel. As this correlator is related to the density of Jordan–Wigner fermions, this behavior is markedly different from the case of free spinless fermions. This difference may be related to the fact that the fermion number is not conserved in the Ising spin chain.

We note that the measurable excess correlations with respect to the equilibrium values are not identically zero outside the classical light cone. Using the asymptotic behavior of the Airy kernel for large positive arguments,

$$K(X, Y) \sim \frac{e^{-\frac{2}{3}(X^{3/2} + Y^{3/2})}}{4\pi(X + Y)}, \qquad (57)$$

we see that their decay is faster than exponential, in accordance with Lieb–Robinson bounds [72].

Corrections to the scaling forms (54), (56) can be obtained by expanding the non-singular part of the integrand to first order in $\tilde{k}$ and $\tilde{q}$. For example, in Eq. (48) we expand

$$F(k,q) = \frac{1}{2}G^{\mathrm{R}}(k,q)e^{i\theta_k}\frac{\varepsilon_k + \varepsilon_q}{4}. \tag{58}$$

The extra terms linear in $\tilde{k}$ and $\tilde{q}$ can be written as derivatives with respect to $n$ and $m$, and eventually lead to the appearance of the partial derivatives of the Airy kernel:

$$\Delta\langle A_n(t)B_m(t)\rangle^{\mathrm{R}} \sim \cos[\kappa(m-n) + \theta_\kappa](1 - 2f_\kappa^{\mathrm{R}})\left(\frac{2}{v_{\max}t}\right)^{1/3}K(X,Y)$$
$$+\left(\frac{2}{v_{\max}t}\right)^{2/3}\frac{e^{i\kappa(m-n)}}{\sin\kappa}[\partial_1 F(\kappa,\kappa)i\partial_X K(X,Y) + \partial_2 F(\kappa,\kappa)(-i)\partial_Y K(X,Y)]$$
$$-\left(\frac{2}{v_{\max}t}\right)^{2/3}\frac{e^{-i\kappa(m-n)}}{\sin\kappa}[\partial_1 F(-\kappa,-\kappa)i\partial_X K(X,Y) + \partial_2 F(-\kappa,-\kappa)(-i)\partial_Y K(X,Y)]. \tag{59}$$

The derivatives of $F(k,q)$ are computed using Eqs. (31) and (173). We note that even these corrections vanish for $\langle A_n B_n \rangle$ if $h < 1$.

Both the scaling forms (54), (56) and the results including these corrections are compared with the numerical data for finite $t$ in Fig. 8 of Appendix E. It is clear that the agreement with the numerical results is improved significantly by the leading order corrections. As we shall see below, they are also responsible for interesting physical phenomena.

### 5.2.2  Critical case

For $h = 1$ the stationary point is at $\kappa = -\pi$. The derivation has to be modified because of the appearance of formally singular terms and because the stationary point is now located at the edge of the Brillouin zone [75]. We shall need the values

$$\varepsilon_\kappa = 2\cos(\kappa/2) = 0, \quad \varepsilon_\kappa''' = -1/4 \neq -v_{\max}, \quad \theta_\kappa = \kappa/2 = -\pi/2, \quad 1 - 2f_\kappa^{\mathrm{L/R}} = 0. \tag{60}$$

Due to the last equality $G^{\mathrm{L/R}} = 0$, so we have to Taylor expand it alongside with $\varepsilon_k$. Moreover, in this case the the second line in Eq. (48) is of the same order as the third line and cannot be dropped. The reason is that both $\varepsilon_k \pm \varepsilon_q$ are zero at $k = q = \kappa$, and out of the otherwise rapidly oscillating factors in the second line $e^{i(\varepsilon_k + \varepsilon_q)t}$ is simply absent for $k = q = \kappa$ while $e^{i(kn+qm)} = e^{-i\pi(n+m)} = e^{-i\pi(n-m)}$ is the same as the analogous factor in the third line. Thus all the four terms in (48) corresponding to the four possibilities $k = \pm\pi$, $q = \pm\pi$ must be taken into account. Because $\pm\pi$ are the edges of the Brillouin zone, after the rescaling (52) the integrals over $K$ and $Q$ are on the positive or negative axis only, again realizing the four possibilities. Luckily, the integrands turn out to be the same in the four terms and their sum can be rewritten as a double integral over the real axis, as in the off-critical case.

Using $\partial_{1,2}G(k,q) = g'(\mp k)$ from Eq. (31) and $g'(\pi) = g'(-\pi)$, the Taylor expansion in all the four terms gives

$$\frac{G^{\mathrm{R/L}}(k,q)e^{i\theta_k}(\varepsilon_k \pm \varepsilon_q)}{\cos(q+i\delta) - \cos(k-i\delta)} = \frac{-ig'(\pi)(\tilde{k}+\tilde{q})^2}{-1/2(\tilde{k}+\tilde{q})(\tilde{k}-\tilde{q}-2i\delta)}, \tag{61}$$

where the $+$ sign is for the $(k,q) \approx (\mp\pi,\pm\pi)$ terms and the $-$ sign is for the $(k,q) \approx (\mp\pi,\mp\pi)$ terms. From here the derivation follows the off-critical case. The surviving $(\tilde{k}+\tilde{q})$ factor translates into derivatives just like in the off-critical corrections, and using $g'(\pi) = -1/(2T)$ we finally obtain

$$\Delta\langle A_n(t)B_m(t)\rangle^{\mathrm{R}} \sim \frac{1}{8T_{\mathrm{R}}}(-1)^{n-m}\left(\frac{8}{t}\right)^{2/3}(\partial_X - \partial_Y)K(X,Y), \tag{62}$$

where $X = (n-t)(8/t)^{1/3}$, $Y = (m-t)(8/t)^{1/3}$. The correction $-i\Delta\langle A_n(t)A_m(t)\rangle^{\mathrm{R}}$ is identical, while the left contributions are opposite in sign.

We found that in the critical case the front is *not* described by the Airy kernel but by its derivative, and the decay is $\sim t^{-2/3}$ instead of $\sim t^{-1/3}$. Moreover, the front in this case is featureless and lacks the typical staircase structure of the Airy kernel (see Fig. 2b).

# 6 Physical observables in the semiclassical limit

In this section we use the general expressions derived in the previous section to calculate the space and time dependence of various physical quantities, including the energy density, energy current, and various correlation functions. The analytical expressions are compared with numerical results obtained by computing the sums in Eq. (26) for finite systems in the the two-temperature case.[8]

## 6.1 Energy density

The energy density in the bulk can be written as (c.f. Eq. (5))

$$h_n = \frac{1}{2}A_{n+1}B_n + \frac{h}{2}A_nB_n. \tag{63}$$

Using the semiclassical result in Eq. (38) we find

$$\langle h_n(t)\rangle_{\mathrm{sc}} = \int_{-\pi}^{\pi} \frac{\mathrm{d}k}{2\pi}\varepsilon_k\left(f_k^{\mathrm{R}}\Theta(x - v_k t) + f_k^{\mathrm{L}}\Theta(-x + v_k t) - \frac{1}{2}\right), \tag{64}$$

where we used the identity (c.f. Eq. (129a) in Appendix A)

$$\cos(-k + \theta_k) + h\cos(\theta_k) = \varepsilon_k. \tag{65}$$

The constant term $-1/2\int_{-\pi}^{\pi}\mathrm{d}k/(2\pi)\varepsilon_k$ corresponds to the ground state (zero point) energy. The rest of the formula has an obvious semiclassical interpretation: in the vicinity of the space-time point $(x, t)$ the energy density is given by the particles arriving at point $x$ at time $t$ coming either from the left or the right with distributions corresponding to the left or right reservoir. In the NESS,

$$\langle h\rangle_{\mathrm{NESS}} = \int_{-\pi}^{\pi} \frac{\mathrm{d}k}{2\pi}\left(\Theta(k)f_k^{\mathrm{R}} + \Theta(-k)f_k^{\mathrm{L}} - \frac{1}{2}\right)\varepsilon_k. \tag{66}$$

The semiclassical formula (64) is compared with the numerical results at different values of $h$ and temperatures in Fig. 1. Expression (64), plotted in dashed line, is seen to capture accurately the actual energy density profiles (dots and solid lines). For lower temperatures, the deviations from the semiclassical result are relatively larger. These corrections are however captured by our analytic results given in Sec. 5.1. The approach to the NESS near the origin is well described by the analytic expression (42), see Fig. 1d. The oscillations, the finite slope and the finite shift are all clearly visible. The numerical data show a longer wavelength modulation as well which is a subleading correction beyond our result.

Near the front the oscillations are enhanced, which is described by the results in Sec. 5.2. Using Eq. (54) in the expression (63) for the energy density and invoking relation (65), we find the scaling behavior for $h \neq 1$ near the right edge

$$\langle h_n(t)\rangle \sim h_{\mathrm{R}} + \varepsilon_\kappa(f_\kappa^{\mathrm{L}} - f_\kappa^{\mathrm{R}})\left(\frac{2}{v_{\max}t}\right)^{1/3}K(X, X), \tag{67}$$

---

[8]In the numerical calculations the edge mode present for $h < 1$ is included.

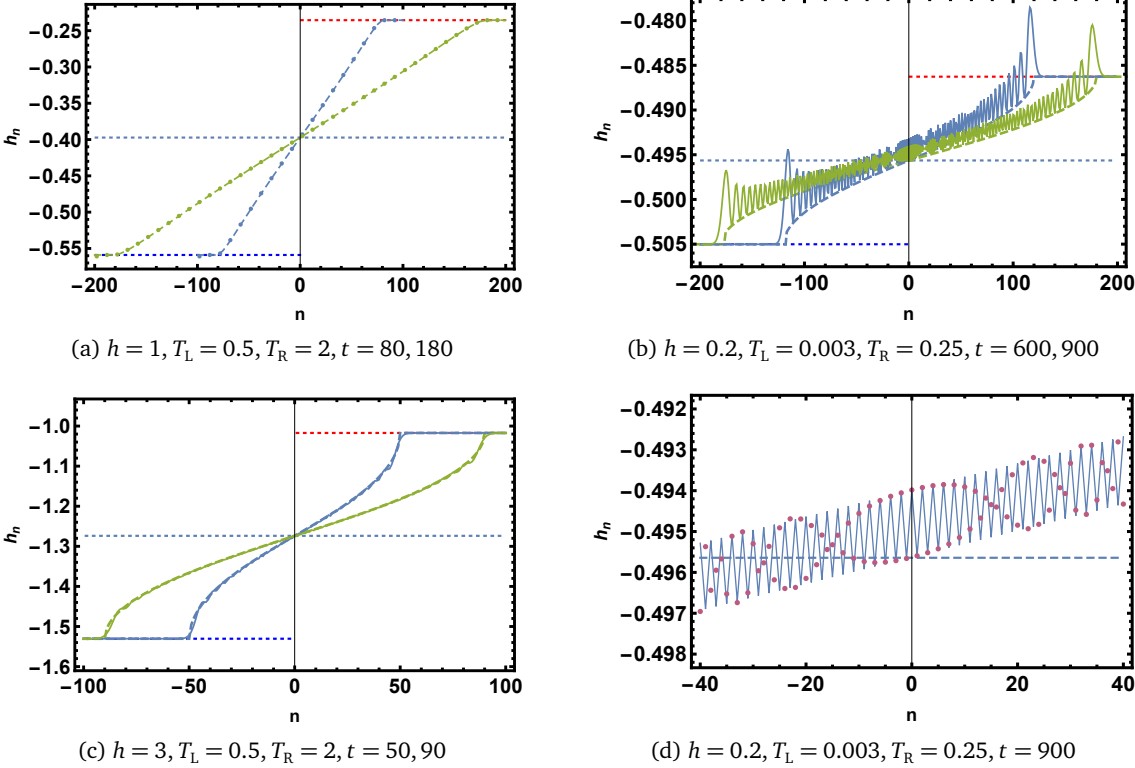

(a) $h = 1, T_L = 0.5, T_R = 2, t = 80, 180$

(b) $h = 0.2, T_L = 0.003, T_R = 0.25, t = 600, 900$

(c) $h = 3, T_L = 0.5, T_R = 2, t = 50, 90$

(d) $h = 0.2, T_L = 0.003, T_R = 0.25, t = 900$

Figure 1: Energy density profiles. (a),(b),(c): Numerical results in solid lines (dots in (a)) are compared with the semiclassical expression (64) shown in dashed lines. The initial thermal energy densities and the NESS value (66) are shown in horizontal dotted lines. (d) Approach to the NESS in the middle, numerical results (dots) are shown together with the analytic result based on Eq. (42) (solid line).

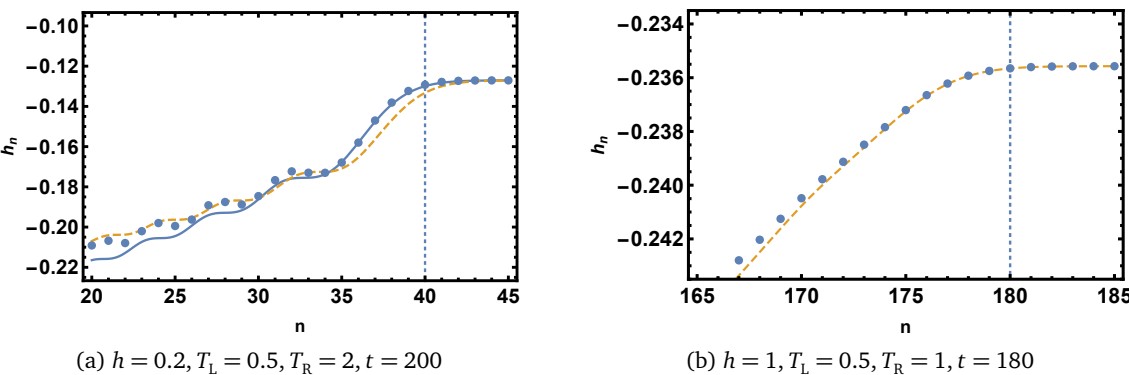

(a) $h = 0.2, T_L = 0.5, T_R = 2, t = 200$

(b) $h = 1, T_L = 0.5, T_R = 1, t = 180$

Figure 2: Behavior of the energy density near the edge of the propagating front. The numerical data (dots) are compared with the scaling results (67) and (68) (dashed line) and with the analytic result involving corrections (solid line). The vertical dotted lines indicate the classical edge of the front set by the maximal group velocity.

where $h_R$ is the bulk energy density on the right and we used that in the large time limit $Y \approx X = [2/(v_{max}t)]^{1/3}(n - v_{max}t)$. This is plotted in Fig. 2a (dashed line) together with numerical results for $h = 0.2$ (dots). For finite time, the corrections (59) involving the derivatives of the Airy kernel and taking into account that $X \neq Y$, shown in solid line in the plots, yield significantly better agreement. It turns out that these corrections are responsible for the

sizeable oscillations in the low temperature case (see Fig. 2b).

In the critical case from Eq. (62) we obtain

$$\langle h_n(t) \rangle \sim h_R + \frac{1}{4t^{2/3}} \left( T_L^{-1} - T_R^{-1} \right) (\partial_X - \partial_Y) K(X, Y) \tag{68}$$

with $X = (n - t)(8/t)^{1/3}, Y = (n + 1 - t)(8/t)^{1/3}$. This is shown in Fig. 2b together with the numerical results. The front is featureless in the critical case and does not have the staircase structure of the Airy kernel.

## 6.2  Energy current

The energy current can be obtained by taking the commutator of the Hamiltonian with the energy density and writing it as a divergence. This yields

$$\mathscr{I}_n^e = \frac{h}{4} (\sigma_n^x \sigma_{n+1}^y - \sigma_n^y \sigma_{n+1}^x) = i \frac{h}{2} (c_n^\dagger c_{n+1} - c_{n+1}^\dagger c_n) = i \frac{h}{4} (A_n A_{n+1} - B_n B_{n+1}). \tag{69}$$

Using Eqs. (37), we obtain the current in the semiclassical limit,

$$\langle \mathscr{I}_n^e(t) \rangle_{sc} = \int_{-\pi}^{\pi} \frac{dk}{2\pi} v_k \varepsilon_k [f_k^R \Theta(x - v_k t) + f_k^L \Theta(-x + v_k t)], \tag{70}$$

where we used that $v_k = -h \sin k / \varepsilon_k$. As $v_k \varepsilon_k$ is the semiclassical contribution of a quasiparticle to the energy current, this result also follows from a semiclassical picture similarly to the energy density: at a given space-time point the energy current is a sum of energy currents carried by single particles that arrive at time $t$ at point $x$. On both sides, the current is zero outside the light cone set by the largest velocity $v_{max}$. It is stratightforward to show that the semiclassical expressions for the energy density and the energy current satisfy the continuity equation,

$$\partial_t \langle h_n(t) \rangle_{sc} + \partial_x \langle \mathscr{I}_n^e(t) \rangle_{sc} = 0. \tag{71}$$

The current in the stationary steady state is

$$\langle \mathscr{I}^e \rangle_{NESS} = \int_{-\pi}^{\pi} \frac{dk}{2\pi} v_k \varepsilon_k [f_k^R \Theta(k) + f_k^L \Theta(-k)], \tag{72}$$

in agreement with the results of Ref. [16]. At the quantum critical point we recover the conformal field theory result [20].

Typical profiles for different values of $h$ and left/right temperatures are shown in Fig. 3. As the right temperature is higher, the current is negative so we plot $-\mathscr{I}_n^e$ instead. In each case, the semiclassical expression (70) (dashed line) agrees well with the numerical results (solid line). In Fig. 3d, the leading order correction based on Eqs. (46) is shown to capture the oscillations near the junction. The amplitudes of the oscillations are relatively large when one of the temperatures is low (c.f. Fig. 3b), so the corrections to the semiclassical limit are more significant. The oscillations are especially pronounced near the edges. Here a surprising phenomenon can also be observed on the hotter side: right after the arrival of the front, the current is initially negative, i.e. it starts to flow in the "wrong" direction, from the colder system to the hot reservoir.

Similarly to the case of the energy density, the edge behavior of the current can be obtained from the results of Sec. 5.2. Combining Eqs. (69) with (56) we find

$$\langle \mathscr{I}_n^e(t) \rangle \sim v_{max} \varepsilon_\kappa (f_\kappa^L - f_\kappa^R) \left( \frac{2}{v_{max} t} \right)^{1/3} K(X, X), \tag{73}$$

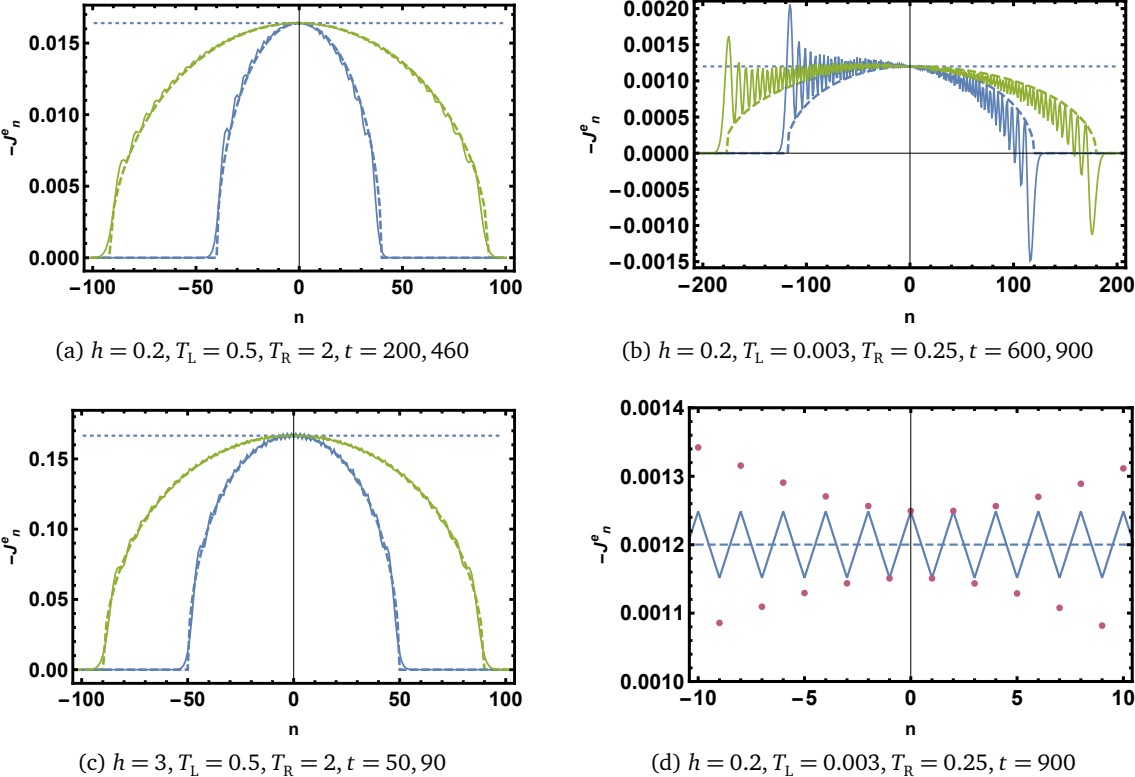

Figure 3: Energy current profiles. (a),(b),(c): Numerical results (solid lines) are compared with the semiclassical expression (70) (dashed lines). The steady state current is shown in horizontal dotted line. In (b) the energy backflow near the right front is clearly visible. (d) Approach to the NESS in the middle, numerical results (dots) are shown together with the analytic result based on Eqs. (46) (solid line).

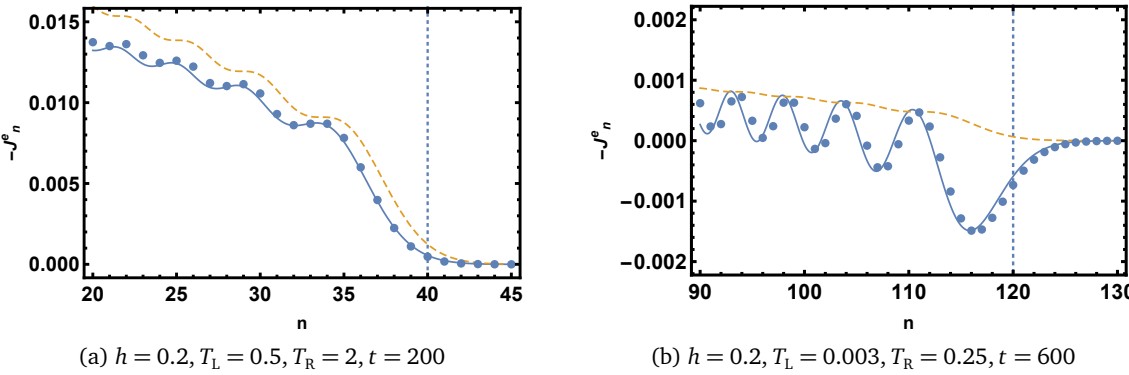

Figure 4: Edge behavior of the energy current near the edge of the propagating front. The numerical data (dots) are compared with the scaling result (73) (dashed line) and with the analytic result involving corrections (solid line). The vertical dotted lines indicate the classical edge of the front set by the maximal propagation velocity.

where we used $v_{max} = -h \sin\kappa / \varepsilon_\kappa$. This result is shown in dashed line in Fig. 4 where it is compared with the numerical data shown in dots. We also plot in solid line the improved result including the first corrections to the scaling form, similar to Eq. (59). We conclude that the corrections to the scaling form (73) are responsible for the large oscillations, and in particular,

for the energy backflow phenomenon.

In the critical case,

$$\langle \mathcal{J}_n^{\mathrm{e}}(t) \rangle \sim \frac{1}{4t^{2/3}} \left( T_{\mathrm{L}}^{-1} - T_{\mathrm{R}}^{-1} \right) (\partial_X - \partial_Y) K(X,Y) \tag{74}$$

with $X = (n-t)(8/t)^{1/3}, Y = (n+1-t)(8/t)^{1/3}$, and the oscillations are absent. The edge behavior of the energy current was first studied in [75] but the result obtained there seems to be different from ours.

## 6.3 Transverse magnetization

The transverse magnetization operator is simply related to the density of the Jordan–Wigner fermions: $\sigma_n^z = 2\sigma_n^+ \sigma_n^- - 1 = 2c_n^\dagger c_n - 1 = -A_n B_n$. Its expectation value in the semiclassical limit is given by (36) for $n = m$:

$$\langle \sigma_n^z(t) \rangle_{\mathrm{sc}} = \int_{-\pi}^{\pi} \frac{dk}{2\pi} \cos \theta_k [(1 - 2f_k^{\mathrm{R}})\Theta(x - v_k t) + (1 - 2f_k^{\mathrm{L}})\Theta(-x + v_k t)]. \tag{75}$$

The $\cos \theta_k$ factor accounts for the Bogoliubov rotation connecting the original fermions with the modes diagonalizing the Hamiltonian. Indeed, using Eqs. (6) and (7), it is straightforward to show that in the $L \to \infty$ limit, in a translationally invariant state

$$-\frac{1}{L} \sum_{n=1}^{L} \langle A_n B_n \rangle = \int_{-\pi}^{\pi} \frac{dk}{2\pi} \cos \theta_k (1 - 2f_k). \tag{76}$$

In the semiclassical picture this correlation is transported through the system by the quasiparticles, which immediately leads to Eq. (75).

The magnetization profile is shown in Fig. 5a for $h = 3, T_{\mathrm{L}} = 0.5, T_{\mathrm{R}} = 2$, for other parameters the profile is similar. The agreement between the semiclassical result (75) and the numerical data is excellent.

The front of the transverse magnetization is described by Eq. (54) and the analogous left contribution:

$$\langle \sigma_n^z(t) \rangle \sim 2 \cos \theta_\kappa (f_\kappa^{\mathrm{R}} - f_\kappa^{\mathrm{L}}) \left( \frac{2}{v_{\mathrm{max}} t} \right)^{1/3} K(X,X) \tag{77}$$

with $X = [2/(v_{\mathrm{max}} t)]^{1/3}(n - v_{\mathrm{max}} t)$. In the light of this result we can revisit the case studied in Ref. [40]. If $T_{\mathrm{L}} = 0, T_{\mathrm{R}} = \infty$, then $f^{\mathrm{L}} = 0$ and $f^{\mathrm{R}} = 1/2$ constants, and we arrive at $\langle \sigma_n^z(t) \rangle \sim \cos \theta_\kappa (v_{\mathrm{max}} t/2)^{-1/3} K(X,X)$. Using the asymptotic expansion of the Airy kernel for large negative arguments, $K(X,X) \approx \sqrt{-X}/\pi$, we recover the square root dependence observed numerically in Ref. [40].

Finally we note that in this simple case (when $T_{\mathrm{L}} = 0, T_{\mathrm{R}} = \infty$ and $f_k^{\mathrm{L}} = 0, f_k^{\mathrm{R}} = 1/2$),

$$\langle \sigma_n^z(t) \rangle_{\mathrm{sc}} = \int_{-\pi}^{\pi} \frac{dk}{2\pi} \cos \theta_k \Theta(-x + v_k t). \tag{78}$$

In the critical case $\varepsilon_k = 2\cos(k/2), v_k = -\sin(k/2), \theta_k = k/2$, and

$$\langle \sigma_n^z(t) \rangle_{\mathrm{sc}} = \int_{-\pi}^{-2\arcsin u} \frac{dk}{2\pi} \cos(k/2) = \frac{1}{\pi}(1 - u), \tag{79}$$

with $u = n/t$, so the profile is linear within the light cone, for $-1 \le u \le 1$ [40]. In this special case the energy density and the current also take simple forms:

$$\langle h_n(t) \rangle_{\mathrm{sc}} = \frac{u-1}{\pi}, \qquad \langle \mathcal{J}_n^{\mathrm{e}}(t) \rangle_{\mathrm{sc}} = \frac{u^2-1}{2\pi}. \tag{80}$$

We note that without the zero point energy the energy density is $\langle h_n(t) \rangle_{\mathrm{sc}} = (u+1)/\pi$.

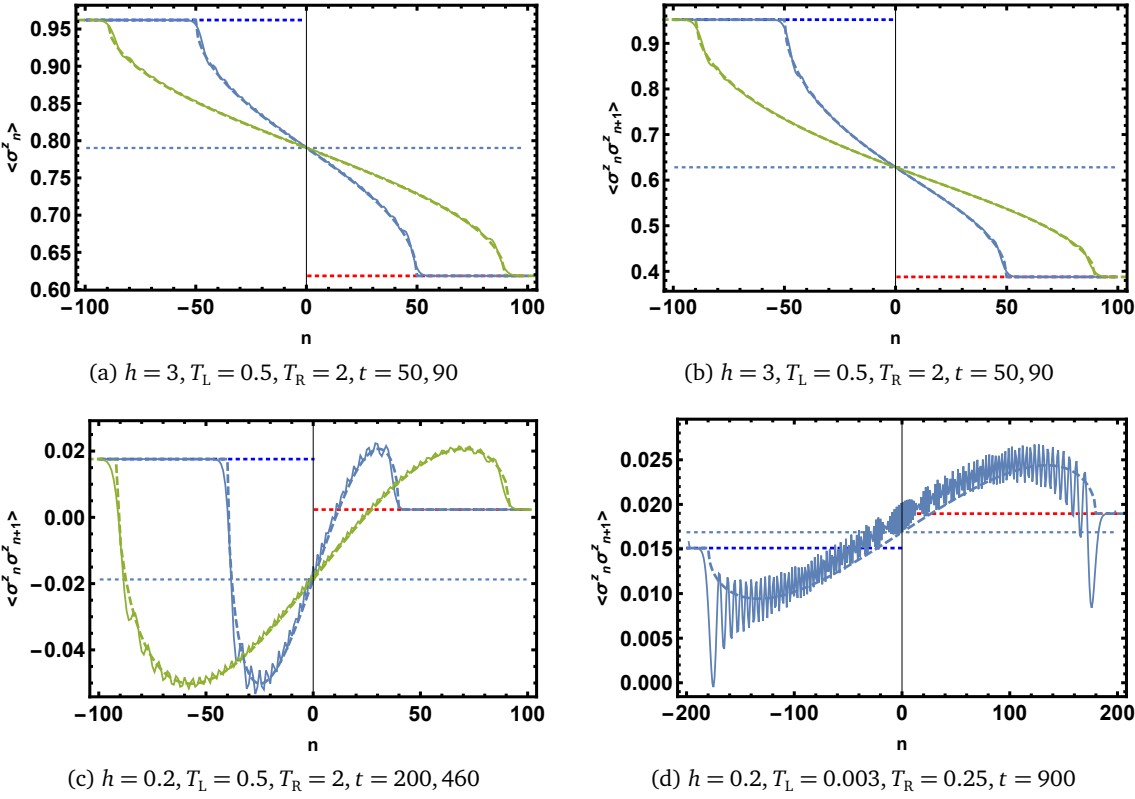

(a) $h = 3, T_{\text{L}} = 0.5, T_{\text{R}} = 2, t = 50, 90$

(b) $h = 3, T_{\text{L}} = 0.5, T_{\text{R}} = 2, t = 50, 90$

(c) $h = 0.2, T_{\text{L}} = 0.5, T_{\text{R}} = 2, t = 200, 460$

(d) $h = 0.2, T_{\text{L}} = 0.003, T_{\text{R}} = 0.25, t = 900$

Figure 5: Transverse magnetization and correlations. Numerical result are shown in solid line while the semiclassical results are plotted in dashed line. The initial thermal values and the NESS values are shown in horizontal dotted lines. (a) Transverse magnetization $\langle \sigma_n^z \rangle$. (b-d) Nearest neighbor transverse spin-spin correlation function $\langle \sigma_n^z \sigma_{n+1}^z \rangle$.

## 6.4 Transverse spin-spin correlations

The transverse spin-spin correlation function is also easy to compute due to the fact that the relation between $\sigma^z$ and the fermion creation/annihilation operator is local. Using Wick's theorem we obtain

$$
\langle \sigma_n^z(t) \sigma_m^z(t) \rangle = \langle A_n(t) B_n(t) A_m(t) B_m(t) \rangle =
$$
$$
\langle A_n(t) B_n(t) \rangle \langle A_m(t) B_m(t) \rangle - \langle A_n(t) A_m(t) \rangle \langle B_n(t) B_m(t) \rangle - \langle A_n(t) B_m(t) \rangle \langle A_m(t) B_n(t) \rangle .
$$
(81)

It is straightforward, albeit lengthy, to write down the semiclassical limit using Eqs. (36),(37). Comparison with numerical data for the nearest neighbor correlator is plotted in Fig. 5b-d. Interestingly, we find that the profile can be very non-monotonic, and the correlator can take values that are very far from the initial equilibrium values on the two half-chains, and can even change sign.

## 6.5 Longitudinal spin-spin correlation function

In contrast to the transverse magnetization, the longitudinal one is non-local in terms of the fermionic degrees of freedom that diagonalize the Hamiltonian. Consequently, correlation functions involving $\sigma_n^x$ are much harder to calculate. In the equal time two-point function, the

Jordan–Wigner string survives only between the two operators so it can be written as

$$\langle \sigma_n^x(t)\sigma_m^x(t)\rangle = \left\langle \prod_{l=n}^{m-1} B_l A_{l+1} \right\rangle. \tag{82}$$

Using Wick's theorem, the multi-point function on the right hand side can be written as the Pfaffian of the matrix

$$\mathbf{\Gamma} = \begin{pmatrix} \Gamma_{n,n} & \Gamma_{n,n+1} & \cdots & \Gamma_{n,m-1} \\ \Gamma_{n+1,n} & \Gamma_{n+1,n+1} & \cdots & \Gamma_{n+1,m-1} \\ \vdots & \vdots & \ddots & \vdots \\ \Gamma_{m-1,n} & \Gamma_{m-1,n+1} & \cdots & \Gamma_{m-1,m-1} \end{pmatrix}, \tag{83}$$

where $\Gamma_{j,l}$ are two-by-two matrices given by

$$\Gamma_{jl} = \begin{pmatrix} \langle B_j B_l\rangle + \delta_{j,l} & \langle B_j A_{l+1}\rangle \\ \langle A_{j+1} B_l\rangle & \langle A_{j+1} A_{l+1}\rangle - \delta_{j,l} \end{pmatrix}. \tag{84}$$

In the semiclassical limit, translational invariance is restored around a given ray. Consequently, $\mathbf{\Gamma}$ becomes a block Toeplitz matrix with blocks

$$\Gamma_{jl} = \begin{pmatrix} -F_{j-l} & G_{j-l} \\ -G_{l-j} & F_{j-l} \end{pmatrix}, \tag{85}$$

where

$$F_n = \int_{-\pi}^{\pi} \frac{dk}{2\pi} e^{ikn}[\Theta(u-v_k)-\Theta(u+v_k)](f_k^{\mathrm{L}}-f_k^{\mathrm{R}}) \equiv \int_{-\pi}^{\pi} \frac{dk}{2\pi} e^{ikn} F(k), \tag{86}$$

$$G_n = \int_{-\pi}^{\pi} \frac{dk}{2\pi} e^{ikn} e^{i\theta_k} e^{-ik}\{[\Theta(u-v_k)+\Theta(u+v_k)](f_k^{\mathrm{L}}-f_k^{\mathrm{R}}) + (1-2f_k^{\mathrm{L}})\} \equiv \int_{-\pi}^{\pi} \frac{dk}{2\pi} e^{ikn} G(k), \tag{87}$$

so the block symbol is

$$\hat{\Gamma}(k) = \begin{pmatrix} -F(k) & G(k) \\ -G(-k) & F(k) \end{pmatrix}. \tag{88}$$

It is of interest to obtain the asymptotic behavior of the correlation function for large separation of the two operators in the semiclassical limit, i.e. when $m-n \gg 1$ but $(m-n)/t \to 0$. First we note that the Pfaffian of an antisymmetric matrix is, up to a sign, the square root of its determinant. The asymptotic behavior of the determinant of a $\ell \times \ell$ block Toeplitz matrix is given by a generalization of the Szegő lemma [73], provided that the matrix satisfies certain properties. Even in thermal equilibrium the Szegő lemma can be directly applied only in the ferromagnetic phase, $h < 1$ [74], so we consider this case only. Generously forgetting about these conditions, a naive application of the lemma gives

$$\log \det \Gamma_\ell \to \ell \int_{-\pi}^{\pi} \frac{dk}{2\pi} \ln \det \hat{\Gamma}(k). \tag{89}$$

This implies that for large separations the correlation function decays exponentially [15],

$$\langle \sigma_n^x \sigma_{n+\ell}^x \rangle \sim e^{-\ell/\xi}, \tag{90}$$

where the correlation length is

$$\xi^{-1} = -\frac{1}{2} \int_{-\pi}^{\pi} \frac{dk}{2\pi} \ln\left[-F(k)^2 + G(k)G(-k)\right]. \tag{91}$$

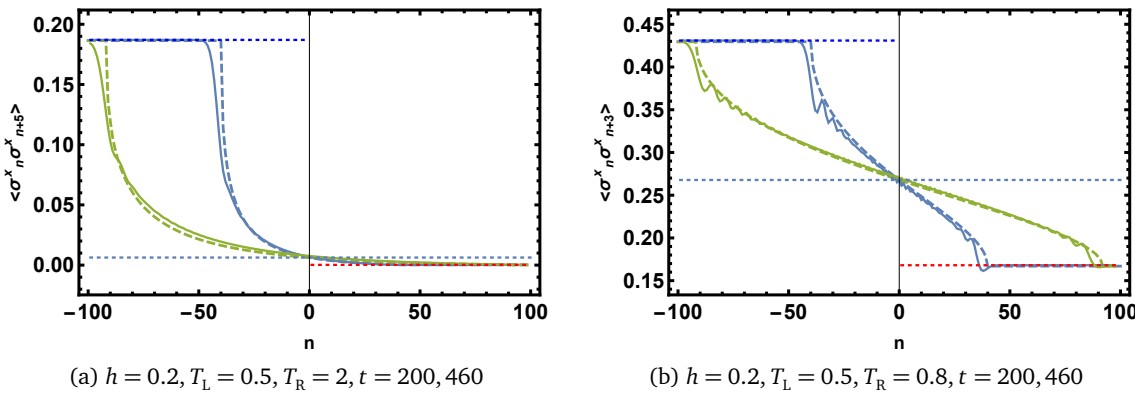

(a) $h = 0.2, T_{\text{L}} = 0.5, T_{\text{R}} = 2, t = 200, 460$       (b) $h = 0.2, T_{\text{L}} = 0.5, T_{\text{R}} = 0.8, t = 200, 460$

Figure 6: The profile of the longitudinal spin-spin correlation function for separation of (a) 5 and (b) 3 lattice spacings. Numerical results (solid line) are compared with Eq. (92) (dashed line). The initial thermal correlations and the NESS value are shown in horizontal dotted lines.

The argument of the logarithm is a long expression, but the integral can be simplified drastically (see Appendix F) and can be shown to be equal to

$$\xi^{-1} = -\int_{-\pi}^{\pi} \frac{dk}{2\pi} \left[ \Theta(v_k + u) \ln(1 - 2f_k^{\text{R}}) + \Theta(v_k - u) \ln(1 - 2f_k^{\text{L}}) \right]. \tag{92}$$

By setting $f_k^{\text{L}} = f_k^{\text{R}}$ we reproduce the space and time independent equilibrium correlation length $\xi^{-1} = -\int_{-\pi}^{\pi} \frac{dk}{2\pi} \ln(1 - 2f_k)$. The inverse correlation length is given by the sum of contributions from particles with left and right momentum distributions. The NESS value, obtained by setting $u = 0$,

$$\xi_{\text{NESS}}^{-1} = -\frac{1}{2} \int_{-\pi}^{\pi} \frac{dk}{2\pi} \ln\left[ \left(1 - 2f_k^{\text{L}}\right)\left(1 - 2f_k^{\text{R}}\right) \right], \tag{93}$$

is simply the average of the left and right inverse correlation lengths.

In Fig. 6 we compare the result of the Szegő lemma, Eqs. (90) and (92), with the profile obtained numerically for two different times. The prefactor of the exponential decay in (90) is set to be the square of the ground state magnetization, $\bar{\sigma}^2 = (1 - h^2)^{1/8}$. The agreement is quite satisfactory despite the fact that (90) is an asymptotic result while we plot it for a separation of a few lattice spacings.

# 7 Domain wall initial state

Even though in the previous section we showed results for the two-temperature scenario, our analytical results apply to a broader class of initial states, namely, to states characterized by fermionic occupations with no anomalous correlations, c.f. Eq. (23). The explicit form of $f(k)$ was only used in the evaluation of some terms appearing in the corrections to the NESS (see Eqs. (172),(177) in Appendix C), but even the formulas (42), (54), (56) remain valid.

As an example, we consider the domain wall or kink initial state where initially the two half chains are fully polarized in the transverse direction: $\langle \sigma_j^z \rangle = +1$ on the left and $\langle \sigma_j^z \rangle = -1$ on the right. An important difference with respect to the two-temperature case is that these initial states are not eigenstates of $H_{\text{L/R}}$, so they have non-trivial time evolution even without joining the two systems. As the semiclassical limit corresponds to $t \to \infty$, the regions outside the light cone have already relaxed by the time the front reaches them. In other words, the scaling

functions for the profiles interpolate between the bulk equilibrium values and not between the initial ones.

Our first task is to compute the initial correlations in momentum space. Using Eqs. (6),

$$\langle \eta_k^\dagger \eta_{k'} \rangle = \frac{1}{4} \sum_{j,l} [\phi_k(j)\phi_{k'}(l)\langle A_j A_l \rangle - \phi_k(j)\psi_{k'}(l)\langle A_j B_l \rangle \psi_k(j)\phi_{k'}(l)\langle B_j A_l \rangle - \psi_k(j)\psi_{k'}(l)\langle B_j B_l \rangle] .$$
(94)

Since $A_j$ and $B_j$ flip spins in the $z$-basis, it is easy to see that

$$\langle A_j A_l \rangle = -\langle B_j B_l \rangle = \delta_{j,l}, \qquad \langle A_j B_l \rangle = -\langle B_j A_l \rangle = \mp \delta_{j,l},$$
(95)

where the upper sign is for the all spins up and the lower sign is for the all spins down state. Using the explicit expressions for the mode functions (7), we obtain in the $L \to \infty$ limit

$$\sum_{j=1}^{L} \phi_k(j)\phi_{k'}(j) = \sum_{j=1}^{L} \psi_k(j)\psi_{k'}(j) = \delta_{k,k'},$$
(96)

$$\sum_{j=1}^{L} \phi_k(j)\psi_{k'}(j) = \sum_{j=1}^{L} \psi_k(j)\phi_{k'}(j) = -\delta_{k,k'}\cos\theta_k .$$
(97)

Plugging this in Eq. (94) we find

$$\langle \eta_k^\dagger \eta_{k'} \rangle = \frac{1}{2}(1 \mp \cos\theta_k)\delta_{k,k'}, \qquad \langle \eta_k \eta_{k'} \rangle = 0,$$
(98)

so the anomalous correlations are zero and $1 - 2f_k = \pm\cos\theta_k$.

Using this in Eq. (64) for the energy density we find

$$\langle h(x,t) \rangle_{\text{sc}} = -\int_{-\pi}^{\pi} \frac{dk}{2\pi} \varepsilon_k/2 \cos\theta_k \left(-\Theta(x - v_k t) + \Theta(-x + v_k t)\right).$$
(99)

Recalling that $\varepsilon_k \cos\theta_k = h + \cos k$ we arrive at

$$\langle h(x,t) \rangle_{\text{sc}} = -\frac{1}{2} \int_{-\pi}^{\pi} \frac{dk}{2\pi} (h + \cos k)\left(-\Theta(x - v_k t) + \Theta(-x + v_k t)\right).$$
(100)

Let $k_1, k_2$ be the solutions of

$$v_{k_{1,2}} = x/t = u$$
(101)

given by

$$k_{1,2} = -\text{sgn}(u)\arccos\left(\frac{-u^2 \mp \sqrt{(1 - u^2)(h^2 - u^2)}}{h}\right).$$
(102)

Then $v_k < u$ implies $-\pi \le k < k_1$ or $k_2 < k \le \pi$ if $u > 0$, and $k_2 < k \le k_1$ if $u < 0$. Then the energy density can be written in the closed form

$$\langle h(x,t) \rangle_{\text{sc}} = \frac{1}{2\pi} \left[h(k_1 - k_2 + \text{sgn}(u)\pi) + \sin k_1 - \sin k_2\right].$$
(103)

In the critical case $\varepsilon_k = 2\cos(k/2)$, $v_k = -\sin(k/2)$, $\theta_k = k/2$, so $k_1 = -\text{sgn}(u)\pi$, $k_2 = -\text{sgn}(u)\arccos(1 - 2u^2) = -2\arcsin u$, which gives

$$\langle h(x,t) \rangle_{\text{sc}} = \frac{1}{\pi}\left(\arcsin u + u\sqrt{1 - u^2}\right).$$
(104)

Similarly, from Eq. (70)

$$\langle \mathscr{I}^{\mathrm{e}}(x,t)\rangle_{\mathrm{sc}} = \int_{-2\arcsin u}^{\pi} \frac{\mathrm{d}k}{2\pi} \sin k \cos k/2 - \int_{-\pi}^{-2\arcsin u} \frac{\mathrm{d}k}{2\pi} \sin k \cos k/2 = -\frac{2}{3\pi}(1-u^2)^{3/2},$$
(105)

and from Eq. (75)

$$\langle \sigma_n^z(t)\rangle_{\mathrm{sc}} = -\int_{-2\arcsin u}^{\pi} \frac{\mathrm{d}k}{2\pi} \cos^2(k/2) + \int_{-\pi}^{-2\arcsin u} \frac{\mathrm{d}k}{2\pi} \cos^2(k/2) = -\frac{1}{\pi}\left(\arcsin u + u\sqrt{1-u^2}\right),$$
(106)

a result first obtained in [36]. The magnetization profile interpolates between the equilibrium values $\pm 1/2$ which are half of the initial magnetizations on the two sides.

# 8  Conclusion

In this paper we studied the out of equilibrium evolution of the transverse field Ising model in the partitioning protocol when the initial state is the tensor product of two states with different macroscopic properties. We focused on the case when initially the anomalous correlations vanish (c.f. Eqs. (23)). We presented several analytical results and performed numerical simulations. Our results are directly applicable to initial states with vanishing anomalous fermionic correlations.

We derived analytic double integral expressions (29),(30) for the fermionic correlators valid for any time and for arbitrary values of the transverse field. Based on these expressions, we obtained the profiles of the fermionic correlators in the space-time scaling limit and showed that the correlations are given by semiclassical expressions (36),(37). As a by-product, we presented a down-to-earth derivation of the correlations in the non-equilibrium steady state.

We also derived the finite time corrections to this current carrying steady state and found a $\sim 1/t$ asymptotic approach for most correlations, although some of them have a $\sim 1/t^2$ approach (see Eqs. (42),(46)). Interestingly, we found that the leading finite time corrections display lattice effects in the off-critical case, which may imply that hydrodynamic approaches based on the local density approximation are not able to capture the leading corrections to the scaling limit, and in particular, they cannot account for sub-ballistic (e.g. diffusive) corrections.

We also investigated the properties of the ballistically propagating front and found that in most off-critical cases it can be described in terms of the Airy kernel. This provides evidence for the universality of the Airy kernel not only for edge phenomena in free fermion systems but also for spin systems that can be mapped to free fermions. It would be interesting to study interacting integrable models in this respect. Comparison with numerical simulations showed the necessity to determine subleading corrections to the universal Airy kernel behavior.

However, perhaps surprisingly and in contrast to the case of free spinless fermions, in the critical case none of the correlations are described the Airy kernel near the edges (see Sec. 5.2.2). The edge behavior is captured by a different kernel given by the derivative of the Airy kernel (see also Ref. [75]). In particular, it is featureless and does not display the typical staircase structure (c.f. Fig. 2b). Moreover, the fermionic density or transverse magnetization is not described by the Airy kernel in the ferromagnetic phase. These findings may be related to the fact that the fermion number is not conserved in the transverse Ising chain.

Based on these general results for the fermionic building blocks, in Section 6 we investigated various physical quantities of the spin chain, including the energy density, the energy current, the transverse magnetization, and two-point correlation functions of the longitudinal and transverse components of the spin. We found that the energy current can show a backflow phenomenon which means that it can flow in the wrong direction (c.f. Figs. 3b and 4a).

This effect is not accounted for by the universal scaling form of the front in terms of the Airy kernel but it is described by the corrections to it involving the derivatives of the kernel. Another interesting finding is the strongly non-monotonic behavior of the transverse spin-spin correlation function (c.f. Fig. 5c).

As for the longitudinal correlation function, we derived a non-rigorous expression (92) for the correlation length based on the naive application of the Szegő lemma. Comparing with numerics, we found very good agreement even for short separations (see Fig. 6).

In the future it would be interesting to study the evolution of dynamical (two-time) spin-spin correlation functions in this and similar setups. The methods of the paper can also be applied to more general inhomogeneous initial states and to situations that involve a global quench as well. As a simple generalization, initial states with non-zero anomalous correlations, $\langle \eta_k \eta_{k'} \rangle \neq 0$, could be considered. The techniques of this work can be naturally carried over to the XY spin chain. The space and time dependent fluctuations and the large deviation functions of different quantities can also be computed, and it would be interesting to check the corresponding extended fluctuation relations [27].

**Note added**    Recently, the work [75] appeared in which some of the results in the space-time scaling limit presented in our paper are derived using a different approach.

## Acknowledgments

The author is grateful to Bruno Bertini, Andrea Gambassi, Viktor Eisler, Gábor Takács, Jacopo Viti, and Zoltán Zimborás for useful discussions.

**Funding information**    The author acknowledges funding from the "Bolyai" Scholarship and a "Prémium" Postdoctoral Grant of the Hungarian Academy of Sciences. He was partially supported by NKFIH grant no. K 119204.

## A   Diagonalizing the Ising chain with open boundary conditions

In this Appendix we provide, in order to be self-contained, details of the diagonalization of the Hamiltonian (3) based on [64, 65]. The Hamiltonian is a bilinear form

$$H = \sum_{i,j=1}^{L} \left[ c_i^\dagger A_{ij} c_j + \frac{1}{2} \left( c_i^\dagger B_{ij} c_j^\dagger - c_i B_{ij} c_j \right) \right] \tag{107}$$

with the real matrices $\mathbf{A}$ and $\mathbf{B}$ given by

$$\mathbf{A} = -\frac{1}{2} \begin{pmatrix} 2h & 1 & 0 & 0 & 0 & \dots & 0 \\ 1 & 2h & 1 & 0 & 0 & \dots & 0 \\ 0 & 1 & 2h & 1 & 0 & \dots & 0 \\ \vdots & & \ddots & \ddots & \ddots & & \vdots \\ 0 & \dots & 0 & 1 & 2h & 1 & 0 \\ 0 & \dots & 0 & 0 & 1 & 2h & 1 \\ 0 & \dots & 0 & 0 & 0 & 1 & 2h \end{pmatrix}, \quad \mathbf{B} = -\frac{1}{2} \begin{pmatrix} 0 & 1 & 0 & 0 & 0 & \dots & 0 \\ -1 & 0 & 1 & 0 & 0 & \dots & 0 \\ 0 & -1 & 0 & 1 & 0 & \dots & 0 \\ \vdots & & \ddots & \ddots & \ddots & & \vdots \\ 0 & \dots & 0 & -1 & 0 & 1 & 0 \\ 0 & \dots & 0 & 0 & -1 & 0 & 1 \\ 0 & \dots & 0 & 0 & 0 & -1 & 0 \end{pmatrix}. \tag{108}$$

It follows that the Hamiltonian can be diagonalized by a linear transformation leading to the modes

$$\eta_k = \sum_{j=1}^{L}\left[g_k(j)c_j + h_k(j)c_j^\dagger\right], \qquad \eta_k^\dagger = \sum_{j=1}^{L}\left[g_k(j)c_j^\dagger + h_k(j)c_j\right]. \tag{109}$$

Here we anticipated that $g_k(j), h_k(j)$ are real. In terms of these modes the Hamiltonian reads

$$H = \sum_k \varepsilon_k \eta_k^\dagger \eta_k + \text{const.} \tag{110}$$

If the $\eta_k$ are canonical fermionic operators (see below), then

$$[\eta_k, H] - \varepsilon_k \eta_k = 0. \tag{111}$$

Substituting Eqs. (107),(109) in this relation, we arrive at

$$\sum_j \left(g_k(j)A_{ji} - h_k(j)B_{ji}\right) = \varepsilon_k g_k(i), \tag{112a}$$

$$\sum_j \left(g_k(j)B_{ji} - h_k(j)A_{ji}\right) = \varepsilon_k h_k(i). \tag{112b}$$

Let us introduce the combinations

$$\phi_k(j) = g_k(j) + h_k(j), \tag{113a}$$
$$\psi_k(j) = g_k(j) - h_k(j), \tag{113b}$$

in terms of which Eqs. (112) read

$$\phi_k(\mathbf{A} - \mathbf{B}) = \varepsilon_k \psi_k, \tag{114a}$$
$$\psi_k(\mathbf{A} + \mathbf{B}) = \varepsilon_k \phi_k, \tag{114b}$$

implying

$$\phi_k(\mathbf{A} - \mathbf{B})(\mathbf{A} + \mathbf{B}) = \varepsilon_k^2 \phi_k, \tag{115a}$$
$$\psi_k(\mathbf{A} + \mathbf{B})(\mathbf{A} - \mathbf{B}) = \varepsilon_k^2 \psi_k. \tag{115b}$$

The product matrices $(\mathbf{A} \pm \mathbf{B})(\mathbf{A} \mp \mathbf{B})$ are real and positive semi-definite, so all the eigenvalues $\varepsilon_k$ are real and it is possible to choose $\phi_k$ and $\psi_k$ to be real and separately orthogonal. If, say, $\phi_k$ is orthonormal, then $\psi_k$ will be orthonormal since

$$\varepsilon_k^2 \psi_k \psi_{k'} = \phi_k(\mathbf{A} - \mathbf{B})(\mathbf{A} + \mathbf{B})\phi_{k'} = \varepsilon_k^2 \phi_k \phi_{k'} = \varepsilon_k^2 \delta_{k,k'}. \tag{116}$$

It is a simple exercise to show that due to the orthogonality of the eigenvectors,

$$\{\eta_k, \eta_{k'}\} = \sum_l [g_k(l)h_{k'}(l) + h_k(l)g_{k'}(l)] = \frac{1}{2}\sum_l [\phi_k(l)\phi_{k'}(l) - \psi_k(l)\psi_{k'}(l)] = 0, \tag{117a}$$

$$\{\eta_k, \eta_{k'}^\dagger\} = \sum_l [g_k(l)g_{k'}(l) + h_k(l)h_{k'}(l)] = \frac{1}{2}\sum_l [\phi_k(l)\phi_{k'}(l) + \psi_k(l)\psi_{k'}(l)] = \delta_{k,k'} \tag{117b}$$

hold, so the transformation (109) is indeed canonical. Using the completeness of the eigenvectors, the inverse transformation can be written as

$$c_j = \sum_k \left[g_k(j)\eta_k + h_k(j)\eta_k^\dagger\right], \qquad c_j^\dagger = \sum_k \left[g_k(j)\eta_k^\dagger + h_k(j)\eta_k\right]. \tag{118}$$

The constant in (110) can be determined exploiting the fact that under a canonical transformation the trace of $H$ is invariant.[9]

It will turn out to be very useful to introduce the combinations

$$A_j = c_j^\dagger + c_j, \qquad B_j = c_j^\dagger - c_j, \tag{119}$$

in terms of which

$$\sigma_j^z = -A_j B_j, \qquad \sigma_j^x \sigma_{j+1}^x = B_j A_{j+1}, \qquad \sigma_j^y \sigma_{j+1}^y = -A_j B_{j+1}. \tag{120}$$

The Jordan–Wigner string is a product of $-\sigma_j^z$ so $\sigma_j^x, \sigma_j^y$ can also be expressed as a product of $A_j$ and $B_j$ operators. Their relation to the mode operators is

$$\eta_k = \frac{1}{2} \sum_{j=1}^{L} \left[ \phi_k(j) A_j - \psi_k(j) B_j \right], \qquad \eta_k^\dagger = \frac{1}{2} \sum_{j=1}^{L} \left[ \phi_k(j) A_j + \psi_k(j) B_j \right], \tag{121}$$

and

$$A_j = \sum_k \phi_k(j)(\eta_k^\dagger + \eta_k), \qquad B_j = \sum_k \psi_k(j)(\eta_k^\dagger - \eta_k). \tag{122}$$

The matrix equations (114) imply

$$h\phi_k(j) + \phi_k(j+1) = -\varepsilon_k \psi_k(j), \qquad 1 \le j < L, \tag{123a}$$
$$h\phi_k(L) = -\varepsilon_k \psi_k(L), \tag{123b}$$

and

$$h\psi_k(1) = -\varepsilon_k \phi_k(1), \tag{124a}$$
$$\psi_k(j-1) + h\psi_k(j) = -\varepsilon_k \phi_k(j), \qquad 1 < j \le L. \tag{124b}$$

These can be translated to

$$h\phi_k(j) + \phi_k(j+1) = -\varepsilon_k \psi_k(j), \qquad 1 \le j \le L, \tag{125a}$$
$$\psi_k(j-1) + h\psi_k(j) = -\varepsilon_k \phi_k(j), \qquad 1 \le j \le L \tag{125b}$$

with boundary conditions

$$\phi(L+1) = 0, \tag{126a}$$
$$\psi(0) = 0. \tag{126b}$$

Plugging in the Ansatz

$$\phi_k(j) = e^{i(kj - \theta_k)}, \tag{127a}$$
$$\psi_k(j) = -e^{ikj}, \tag{127b}$$

we obtain

$$e^{ik} + h = \varepsilon_k e^{i\theta_k}, \tag{128a}$$
$$e^{-ik} + h = \varepsilon_k e^{-i\theta_k}, \tag{128b}$$

---

[9]The notation is hiding the fact that the problem originally involved $2L \times 2L$ matrices acting on vectors with entries $c_j$ and $c_j^\dagger$. This is reflected by the fact that we have two sets of $L$ basis vectors. In principle we need $2L$ vectors to diagonalize the Hamiltonian. But notice that Eqs. (114) have the symmetry $\varepsilon_k \to -\varepsilon_k$, $\psi_k \to -\psi_k$. This amounts to interchanging $g_k$ and $h_k$ or $\eta_k$ with $\eta_k^\dagger$, thus it is a particle-hole transformation, which renders the state with all negative energy states filled the new vaccum.

or taking the real and imaginary parts,

$$h + \cos k = \varepsilon_k \cos \theta_k \,, \tag{129a}$$

$$\sin k = \varepsilon_k \sin \theta_k \,, \tag{129b}$$

which gives for the Bogoliubov angle

$$\tan \theta_k = \frac{\sin k}{h + \cos k} \,. \tag{130}$$

As we will see, $k \in [0, \pi]$, so if we set $\theta_k \in [0, \pi]$ then $\varepsilon_k \geq 0$. Moreover, $\sin \theta_k > 0$ and $\tan \theta_k, \cos \theta_k$ are negative when $h + \cos k < 0$. For the energy eigenvalues we get

$$\varepsilon_k = \sqrt{(h + \cos k)^2 + \sin k^2} = \sqrt{1 + 2h \cos k + h^2} \,. \tag{131}$$

Imposing the boundary condition for $\psi_k$ we find

$$\phi_k(j) = A_k \sin(kj - \theta_k) \,, \tag{132a}$$

$$\psi_k(j) = -A_k \sin(kj) \,. \tag{132b}$$

Imposing the boundary condition for $\phi_k$ leads to the quantization condition

$$k(L + 1) - \theta_k = n\pi \,, \qquad n \in \mathbb{Z} \,, \tag{133}$$

or

$$k = \frac{\pi}{L + 1} \left[ n + \frac{1}{\pi} \arctan\left( \frac{\sin k}{h + \cos k} \right) \right] \,. \tag{134}$$

Note that this means that

$$\phi_k(j) = -A_k(-1)^n \sin[k(L + 1 - j)] \,, \tag{135a}$$

$$\psi_k(j) = -A_k \sin(kj) \,. \tag{135b}$$

The normalization constant is

$$A_k^{-2} = \sum_{j=1}^{L} \sin^2(kj) = \sum_{j=1}^{L} \sin^2(kj - \theta_k) =$$

$$\frac{L}{2} - \frac{\cos[k(L+1)]\sin(kL)}{2\sin k} = \frac{L}{2} + h\frac{\sin[2k(L+1)]}{4\sin k} = \frac{L}{2} + h\frac{\cos\theta_k}{2\varepsilon_k} = \frac{L}{2} + \frac{h(h + \cos k)}{2\varepsilon_k^2} \,. \tag{136}$$

Let us collect some useful relations:

$$\sin[(L + 1)k] = (-1)^n \sin \theta_k = (-1)^n \frac{\sin k}{\varepsilon_k} \,, \tag{137}$$

$$\sin(Lk) = (-1)^n \sin(\theta_k - k) = (-1)^{n+1} \frac{h \sin k}{\varepsilon_k} \,, \tag{138}$$

so $\varepsilon_k = \sin k / \sin \theta_k$ and the quantization condition can be written as

$$\frac{\sin[(L + 1)k]}{\sin(Lk)} = -\frac{1}{h} \,. \tag{139}$$

For $h > 1$ there are always $L$ real solutions, while for $h < 1$ it is possible to have a complex root corresponding to a localized mode. To see this let us consider the last quantum number,

$n = L$. The value $k = \pi - 0$ is always a solution (remember that $0 < \theta_k < \pi$,) but it is unphysical. Expanding Eq. (9) around $k = \pi$ we find the solution

$$k \approx \pi - |1 - h|\sqrt{6\frac{h - L(1 - h)}{h(h + 1)}} \tag{140}$$

which is real if

$$h > \frac{L}{L + 1}. \tag{141}$$

This is always satisfied for $h > 1$, but for $h < 1$ it is true only if $L < h/(1 - h)$. So for a large enough $L$ at fixed $h$ there will always be a complex solution of the form $k_0 = \pi + i\nu$ with the imaginary part satisfying

$$\nu = -\frac{1}{L + 1}\text{arctanh}\left(\frac{\sinh\nu}{h - \cosh\nu}\right). \tag{142}$$

For $L$ large, $\nu$ quickly approaches $-\ln h$. Expanding around it we find

$$\nu = -\ln h - (1 - h^2)h^{2L} + \mathcal{O}(Lh^{4L}). \tag{143}$$

The energy eigenvalue is exponentially small:

$$\varepsilon_{k_0} = (1 - h^2)h^L + \mathcal{O}(h^{2L}). \tag{144}$$

The corresponding eigenvectors are

$$\phi_{k_0}(j) = A_{k_0}(-1)^j \sinh[(L + 1 - j)\nu], \tag{145a}$$

$$\psi_{k_0}(j) = -A_{k_0}(-1)^j \sinh(\nu j), \tag{145b}$$

which shows that these modes are localized at the edges of the chain. The normalization factor behaves for large $L$ as $A_{k_0} \approx 2\sqrt{1 - h^2}h^L$.

At criticality $h = 1$, $\theta_k = k/2$, and the quantization condition becomes

$$k = \frac{2n\pi}{2L + 1}, \tag{146}$$

The eigenfunctions are

$$\phi_k(j) = \frac{2}{\sqrt{2L + 1}}\sin[k(j - 1/2)], \tag{147a}$$

$$\psi_k(j) = -\frac{2}{\sqrt{2L + 1}}\sin(kj) \tag{147b}$$

with their energy being $\varepsilon_k = 2\cos\frac{k}{2}$.

# B  Exact time evolved fermionic correlations

In this Appendix we explain how to arrive at the exact double integral expressions for the fermionic correlation functions.

First note that the contributions of the left and right half-chains can be related to each other. For example, using Eq. (16),

$$
\begin{aligned}
\langle A_{1-m}(t)B_{1-n}(t)\rangle^{\mathrm{R}} = \\
\sum_{j,l=1}^{\infty} \Big[ \langle A_j|A_{1-m}(t)\rangle\langle B_l|B_{1-n}(t)\rangle - \langle A_j|B_{1-n}(t)\rangle\langle B_l|A_{1-m}(t)\rangle \Big]\langle A_{1-l}B_{1-j}\rangle_0|_{f^{\mathrm{L}}\to f^{\mathrm{R}}} \\
= \sum_{j',l'=-\infty}^{0} \Big[ \langle B_{l'}|B_m(t)\rangle\langle A_{j'}|A_n(t)\rangle - \langle B_{l'}|A_n(t)\rangle\langle A_{j'}|B_m(t)\rangle \Big]\langle A_{j'}B_{l'}\rangle_0|_{f^{\mathrm{L}}\to f^{\mathrm{R}}} \\
= \langle A_n(t)B_m(t)\rangle^{\mathrm{L}}|_{f^{\mathrm{L}}\to f^{\mathrm{R}}}, \quad (148)
\end{aligned}
$$

where we changed variables to $\tilde{j} = 1-l, \tilde{l} = 1-j$, and used $\langle X_{1-n}|Y_{1-m}(t)\rangle = \langle Y_n|X_m(t)\rangle$ and $\langle A_j|A_n(t)\rangle = \langle B_j|B_n(t)\rangle$ which follow from Eqs. (20).
Similarly we can derive $\langle A_n(t)A_m(t)\rangle^{\mathrm{R}} = \langle B_{1-m}(t)B_{1-n}(t)\rangle^{\mathrm{L}}|_{f^{\mathrm{L}}\to f^{\mathrm{R}}}$.

We now turn to the manipulation of the correlation functions. Our starting point is the exact expression (26) with (27). Substituting the coefficients from Eqs. (20) and the initial correlations from Eqs. (25) we obtain

$$
\langle A_n(t)A_m(t)\rangle = \delta_{n,m} + \sum_{v=\mathrm{L},\mathrm{R}}\int_{-\pi}^{\pi}\frac{\mathrm{d}q}{\pi}(1-2f_q^v)\Big[\Phi_q^v(n,t)\Psi_q^v(m,t) - \Phi_q^v(m,t)\Psi_q^v(n,t)\Big], \quad (149\mathrm{a})
$$

$$
\langle B_n(t)B_m(t)\rangle = -\delta_{n,m} + \sum_{v=\mathrm{L},\mathrm{R}}\int_{-\pi}^{\pi}\frac{\mathrm{d}q}{\pi}(1-2f_q^v)\Big[Z_q^v(n,t)\Xi_q^v(m,t) - Z_q^v(m,t)\Xi_q^v(n,t)\Big],
$$
$$(149\mathrm{b})$$

$$
\langle A_n(t)B_m(t)\rangle = \sum_{v=\mathrm{L},\mathrm{R}}\int_{-\pi}^{\pi}\frac{\mathrm{d}q}{\pi}(1-2f_q^v)\Big[\Phi_q^v(n,t)\Xi_q^v(m,t) - Z_q^v(m,t)\Psi_q^v(n,t)\Big]. \quad (149\mathrm{c})
$$

Here we introduced

$$
\Phi_q^v(n,t) = \int_{-\pi}^{\pi}\frac{\mathrm{d}k}{2\pi}\tilde{\varphi}_k(n)\cos(\varepsilon_k t)\mathscr{A}_{k,q}^v, \quad (150\mathrm{a})
$$

$$
\Psi_q^v(n,t) = i\int_{-\pi}^{\pi}\frac{\mathrm{d}k}{2\pi}\tilde{\varphi}_k(n)\sin(\varepsilon_k t)\mathscr{B}_{k,q}^v, \quad (150\mathrm{b})
$$

$$
\Xi_q^v(n,t) = \int_{-\pi}^{\pi}\frac{\mathrm{d}k}{2\pi}\tilde{\varphi}_k(n)\cos(\varepsilon_k t)\mathscr{C}_{k,q}^v, \quad (150\mathrm{c})
$$

$$
Z_q^v(n,t) = i\int_{-\pi}^{\pi}\frac{\mathrm{d}k}{2\pi}\tilde{\chi}_k(n)\sin(\varepsilon_k t)\mathscr{A}_{k,q}^v. \quad (150\mathrm{d})
$$

They can be interpreted as time evolved eigenfunctions with

$$
\mathscr{A}_{k,q}^v = \sum_{j\in v}\tilde{\varphi}_k^*(j)\tilde{\phi}_q^v(j), \quad (151\mathrm{a})
$$

$$
\mathscr{B}_{k,q}^v = \sum_{j\in v}\tilde{\chi}_k^*(j)\tilde{\psi}_q^v(j), \quad (151\mathrm{b})
$$

$$
\mathscr{C}_{k,q}^v = \sum_{j\in v}\tilde{\varphi}_k^*(j)\tilde{\psi}_q^v(j) \quad (151\mathrm{c})
$$

being the overlaps between the eigenmodes of the half systems and those of the full system.

Explicitly,

$$
\mathscr{A}_{k,q}^{\mathrm{R}} = \sum_{j=1}^{\infty} e^{ikj-i\theta_k} \sin(qj - \theta_q) = e^{-i\theta_k} \frac{1}{2} \frac{\sin(q - \theta_q) + e^{ik}\sin\theta_q}{\cos(k+i\delta) - \cos q} = \frac{1}{2} \frac{e^{-i\theta_k}(h + e^{ik})\sin q/\varepsilon_q}{\cos(k+i\delta) - \cos q}
$$

$$
= \frac{1}{2} \frac{\varepsilon_k}{\varepsilon_q} \frac{\sin q}{\cos(k+i\delta) - \cos q}, \tag{152a}
$$

$$
\mathscr{B}_{k,q}^{\mathrm{R}} = \sum_{j=1}^{\infty} e^{ikj} \sin(qj) = \frac{1}{2} \frac{\sin q}{\cos(k+i\delta) - \cos q}, \tag{152b}
$$

$$
\mathscr{C}_{k,q}^{\mathrm{R}} = -\sum_{j=1}^{\infty} e^{ikj-i\theta_k} \sin(qj) = -\frac{1}{2} \frac{e^{-i\theta_k}\sin q}{\cos(k+i\delta) - \cos q}, \tag{152c}
$$

$$
\mathscr{A}_{k,q}^{\mathrm{L}} = -\sum_{j=-\infty}^{0} e^{ikj-i\theta_k} \sin(q(1-j)) = -\frac{1}{2} \frac{e^{ik-i\theta_k}\sin q}{\cos(k-i\delta) - \cos q}, \tag{152d}
$$

$$
\mathscr{B}_{k,q}^{\mathrm{L}} = -\sum_{j=-\infty}^{0} e^{ikj} \sin(q(1-j) - \theta_q) = -\frac{1}{2} \frac{\varepsilon_k}{\varepsilon_q} e^{-i\theta_k} e^{ik} \frac{\sin q}{\cos(k-i\delta) - \cos q}, \tag{152e}
$$

$$
\mathscr{C}_{k,q}^{\mathrm{L}} = \sum_{j=-\infty}^{0} e^{ikj-i\theta_k} \sin(q(1-j) - \theta_q) = \frac{1}{2} \frac{\varepsilon_k}{\varepsilon_q} e^{-2i\theta_k} e^{ik} \frac{\sin q}{\cos(k-i\delta) - \cos q}. \tag{152f}
$$

Let us list all the products of the time dependent eigenfunctions that appear in Eqs. (149):

$$
\Phi^{\mathrm{R}}(n,t)\Psi^{\mathrm{R}}(m,t) =
$$
$$
\int_{-\pi}^{\pi} \frac{\mathrm{d}k}{2\pi} \int_{-\pi}^{\pi} \frac{\mathrm{d}k'}{2\pi} \check{\varphi}_k(n) \check{\varphi}_{k'}(m) \cos(\varepsilon_k t) i \sin(\varepsilon_{k'} t) \frac{1}{4} \frac{\varepsilon_k}{\varepsilon_q} \frac{\sin^2 q}{[\cos(k+i\delta) - \cos q][\cos(k'+i\delta) - \cos q]}, \tag{153a}
$$

$$
\Phi^{\mathrm{L}}(n,t)\Psi^{\mathrm{L}}(m,t) =
$$
$$
\int_{-\pi}^{\pi} \frac{\mathrm{d}k}{2\pi} \int_{-\pi}^{\pi} \frac{\mathrm{d}k'}{2\pi} \check{\varphi}_k(n) \check{\varphi}_{k'}(m) \cos(\varepsilon_k t) i \sin(\varepsilon_{k'} t) \frac{1}{4} \frac{\varepsilon_{k'}}{\varepsilon_q} \frac{e^{i(k+k')-i(\theta_k+\theta_{k'})}\sin^2 q}{[\cos(k-i\delta) - \cos q][\cos(k'-i\delta) - \cos q]}, \tag{153b}
$$

$$
Z^{\mathrm{R}}(n,t)\Xi^{\mathrm{R}}(m,t) =
$$
$$
\int_{-\pi}^{\pi} \frac{\mathrm{d}k}{2\pi} \int_{-\pi}^{\pi} \frac{\mathrm{d}k'}{2\pi} \check{\chi}_k(n) \check{\varphi}_{k'}(m) i \sin(\varepsilon_k t) \cos(\varepsilon_{k'} t) (-\frac{1}{4}) \frac{\varepsilon_k}{\varepsilon_q} \frac{e^{-i\theta_{k'}}\sin^2 q}{[\cos(k+i\delta) - \cos q][\cos(k'+i\delta) - \cos q]}, \tag{153c}
$$

$$
Z^{\mathrm{L}}(n,t)\Xi^{\mathrm{L}}(m,t) =
$$
$$
\int_{-\pi}^{\pi} \frac{\mathrm{d}k}{2\pi} \int_{-\pi}^{\pi} \frac{\mathrm{d}k'}{2\pi} \check{\chi}_k(n) \check{\varphi}_{k'}(m) \cos(\varepsilon_k t) \cos(\varepsilon_{k'} t) (-\frac{1}{4}) \frac{\varepsilon_{k'}}{\varepsilon_q} \frac{e^{i(k+k')-i(\theta_k+2\theta_{k'})}\sin^2 q}{[\cos(k-i\delta) - \cos q][\cos(k'-i\delta) - \cos q]}, \tag{153d}
$$

$$
\Phi^{\mathrm{R}}(n,t)\Xi^{\mathrm{R}}(m,t) =
$$
$$
\int_{-\pi}^{\pi} \frac{\mathrm{d}k}{2\pi} \int_{-\pi}^{\pi} \frac{\mathrm{d}k'}{2\pi} \check{\varphi}_k(n) \check{\varphi}_{k'}(m) \cos(\varepsilon_k t) \cos(\varepsilon_{k'} t) (-\frac{1}{4}) \frac{\varepsilon_k}{\varepsilon_q} \frac{e^{-i\theta_{k'}}\sin^2 q}{[\cos(k+i\delta) - \cos q][\cos(k'+i\delta) - \cos q]}, \tag{153e}
$$

$$\Phi^{\mathrm{L}}(n,t)\Xi^{\mathrm{L}}(m,t)=$$
$$\int_{-\pi}^{\pi}\frac{\mathrm{d}k}{2\pi}\int_{-\pi}^{\pi}\frac{\mathrm{d}k'}{2\pi}\tilde{\varphi}_k(n)\tilde{\varphi}_{k'}(m)\cos(\varepsilon_k t)\cos(\varepsilon_{k'}t)(-\frac{1}{4})\frac{\varepsilon_{k'}}{\varepsilon_q}\frac{e^{i(k+k')-i(\theta_k+2\theta_{k'})}\sin^2 q}{[\cos(k-i\delta)-\cos q][\cos(k'-i\delta)-\cos q]},$$
$$(153\mathrm{f})$$

$$Z^{\mathrm{R}}(n,t)\Psi^{\mathrm{R}}(m,t)=$$
$$-\int_{-\pi}^{\pi}\frac{\mathrm{d}k}{2\pi}\int_{-\pi}^{\pi}\frac{\mathrm{d}k'}{2\pi}\tilde{\chi}_k(n)\tilde{\varphi}_{k'}(m)\sin(\varepsilon_k t)\sin(\varepsilon_{k'}t)\frac{1}{4}\frac{\varepsilon_k}{\varepsilon_q}\frac{\sin^2 q}{[\cos(k+i\delta)-\cos q][\cos(k'+i\delta)-\cos q]},$$
$$(153\mathrm{g})$$

$$Z^{\mathrm{L}}(n,t)\Psi^{\mathrm{L}}(m,t)=$$
$$-\int_{-\pi}^{\pi}\frac{\mathrm{d}k}{2\pi}\int_{-\pi}^{\pi}\frac{\mathrm{d}k'}{2\pi}\tilde{\varphi}_k(n)\tilde{\chi}_{k'}(m)\sin(\varepsilon_k t)\sin(\varepsilon_{k'}t)\frac{1}{4}\frac{\varepsilon_{k'}}{\varepsilon_q}\frac{e^{i(k+k')-i(\theta_k+\theta_{k'})}\sin^2 q}{[\cos(k-i\delta)-\cos q][\cos(k'-i\delta)-\cos q]}.$$
$$(153\mathrm{h})$$

Interchanging the integral over $q$ with the integrals over $k,k'$ in Eqs. (149) we see that for all correlation functions we need to evaluate the integral

$$Q^\nu(k,k')=\int_{-\pi}^{\pi}\frac{\mathrm{d}q}{\pi}(1-2f_q^\nu)\frac{1}{4\varepsilon_q}\frac{\sin^2 q}{(\cos(k+i\delta)-\cos q)(\cos(k'-i\delta)-\cos q)}$$
$$=-\frac{1}{\cos(k'-i\delta)-\cos(k+i\delta)}\int_{-\pi}^{\pi}\frac{\mathrm{d}q}{\pi}(1-2f_q^\nu)\frac{\sin^2 q}{4\varepsilon_q}\left(\frac{1}{\cos q-\cos(k+i\delta)}-\frac{1}{\cos q-\cos(k'-i\delta)}\right)$$
$$=-\frac{1}{\cos(k'-i\delta)-\cos(k+i\delta)}\frac{1}{2i}[g^\nu(k)-g^\nu(-k)],\quad(154)$$

where $\nu$ stands for R,L and we introduced

$$g^\nu(k)=2i\int_{-\pi}^{\pi}\frac{\mathrm{d}q}{\pi}(1-2f_q^\nu)\frac{\sin^2 q}{4\varepsilon_q}\frac{1}{\cos q-\cos(k+i\delta)}.\qquad(155)$$

Changing variables to $z=e^{iq}$, the integral becomes a contour integral on the unit circle going around the origin:

$$g(k)=2i\oint\frac{\mathrm{d}z}{i\pi z}\frac{-(1-2f_{q(z)})(z-z^{-1})^2}{16\sqrt{(h+z)(h+z^{-1})}}\frac{2z}{(z-e^{ik-\delta})(z-e^{-ik+\delta})}.\qquad(156)$$

Now we separate the contributions of the simple pole at $z=e^{ik-\delta}$:

$$g(k)=\frac{\sin k}{\varepsilon_k}(1-2f_k)+I(k),\qquad(157)$$

where

$$I(k)=-\oint_{\mathscr{C}}\frac{\mathrm{d}z}{4\pi}\frac{(1-2f_{q(z)})(z-z^{-1})^2}{\sqrt{(h+z)(h+z^{-1})}}\frac{1}{(z-e^{ik})(z-e^{-ik})}\qquad(158)$$

is a contour integral around a circle $\mathscr{C}$ of radius $\min(h,h^{-1})<r<1$. We will not evaluate the integral, only note that it is an even function of $k$.

Now we are in the position to write down fermionic correlations. Introducing the notation

$$G^\nu(k,k')=g^\nu(k)-g^\nu(-k'),\qquad(159)$$

the contribution of the right half to the correlators are

$$\left[\langle A_n(t)A_m(t)\rangle - \delta_{n,m}\right]_{\mathrm{R}} =$$
$$-\frac{1}{2}\int_{-\pi}^{\pi}\frac{dk}{2\pi}\int_{-\pi}^{\pi}\frac{dk'}{2\pi}G^{\mathrm{R}}(k,k')\varepsilon_k e^{i(\theta_k-\theta_{k'})}\cos(\varepsilon_k t)\sin(\varepsilon_{k'}t)\frac{e^{i(k'm-kn)}-e^{i(k'n-km)}}{\cos(k'-i\delta)-\cos(k+i\delta)}$$
$$=-\frac{1}{2}\int_{-\pi}^{\pi}\frac{dk}{2\pi}\int_{-\pi}^{\pi}\frac{dk'}{2\pi}G^{\mathrm{R}}(k,k')e^{i(\theta_k-\theta_{k'})}\frac{\varepsilon_k\sin(\varepsilon_{k'}t)\cos(\varepsilon_k t)-\varepsilon_{k'}\sin(\varepsilon_k t)\cos(\varepsilon_{k'}t)}{\cos(k'-i\delta)-\cos(k+i\delta)}e^{i(k'm-kn)},$$
$$(160)$$

where we made the variable change $k \to -k'$, $k' \to -k$ for the second exponential in the numerator. Similarly,

$$\left[\langle B_n(t)B_m(t)\rangle + \delta_{n,m}\right]_{\mathrm{R}} =$$
$$-\frac{1}{2}\int_{-\pi}^{\pi}\frac{dk}{2\pi}\int_{-\pi}^{\pi}\frac{dk'}{2\pi}G^{\mathrm{R}}(k,k')\frac{\varepsilon_k\sin(\varepsilon_k t)\cos(\varepsilon_{k'}t)-\varepsilon_{k'}\sin(\varepsilon_{k'}t)\cos(\varepsilon_k t)}{\cos(k'-i\delta)-\cos(k+i\delta)}e^{i(k'm-kn)},\quad(161)$$

and

$$\left[\langle A_n(t)B_m(t)\rangle\right]_{\mathrm{R}} =$$
$$-\frac{i}{2}\int_{-\pi}^{\pi}\frac{dk}{2\pi}\int_{-\pi}^{\pi}\frac{dk'}{2\pi}G^{\mathrm{R}}(k,k')e^{i\theta_k}\frac{\varepsilon_k\cos(\varepsilon_k t)\cos(\varepsilon_{k'}t)+\varepsilon_{k'}\sin(\varepsilon_k t)\sin(\varepsilon_{k'}t)}{\cos(k'-i\delta)-\cos(k+i\delta)}e^{i(k'm-kn)}.$$
$$(162)$$

The final result is obtained by adding the left and right contributions:

$$\langle A_n(t)A_m(t)\rangle - \delta_{n,m} =$$
$$-\frac{1}{2}\int_{-\pi}^{\pi}\frac{dk}{2\pi}\int_{-\pi}^{\pi}\frac{dk'}{2\pi}G^{\mathrm{R}}(k,k')e^{i(\theta_k-\theta_{k'})}\frac{\varepsilon_k\cos(\varepsilon_k t)\sin(\varepsilon_{k'}t)-\varepsilon_{k'}\sin(\varepsilon_k t)\cos(\varepsilon_{k'}t)}{\cos(k'-i\delta)-\cos(k+i\delta)}e^{i(k'm-kn)}$$
$$-\frac{1}{2}\int_{-\pi}^{\pi}\frac{dk}{2\pi}\int_{-\pi}^{\pi}\frac{dk'}{2\pi}G^{\mathrm{L}}(k,k')e^{i(k'-k)}\frac{\varepsilon_k\sin(\varepsilon_k t)\cos(\varepsilon_{k'}t)-\varepsilon_{k'}\cos(\varepsilon_k t)\sin(\varepsilon_{k'}t)}{\cos(k'-i\delta)-\cos(k+i\delta)}e^{i(km-k'n)},$$
$$(163)$$

$$\langle B_n(t)B_m(t)\rangle + \delta_{n,m} =$$
$$-\frac{1}{2}\int_{-\pi}^{\pi}\frac{dk}{2\pi}\int_{-\pi}^{\pi}\frac{dk'}{2\pi}G^{\mathrm{R}}(k,k')\frac{\varepsilon_k\sin(\varepsilon_k t)\cos(\varepsilon_{k'}t)-\varepsilon_{k'}\cos(\varepsilon_k t)\sin(\varepsilon_{k'}t)}{\cos(k'-i\delta)-\cos(k+i\delta)}e^{i(k'm-kn)}$$
$$-\frac{1}{2}\int_{-\pi}^{\pi}\frac{dk}{2\pi}\int_{-\pi}^{\pi}\frac{dk'}{2\pi}G^{\mathrm{L}}(k,k')e^{i(\theta_k-\theta_{k'})}\frac{\varepsilon_k\cos(\varepsilon_k t)\sin(\varepsilon_{k'}t)-\varepsilon_{k'}\sin(\varepsilon_k t)\cos(\varepsilon_{k'}t)}{\cos(k'-i\delta)-\cos(k+i\delta)}e^{i(km-k'n)}e^{i(k'-k)},$$
$$(164)$$

$$\langle A_n(t)B_m(t)\rangle = -\langle B_m(t)A_m(t)\rangle =$$
$$-\frac{i}{2}\int_{-\pi}^{\pi}\frac{dk}{2\pi}\int_{-\pi}^{\pi}\frac{dk'}{2\pi}G^{\mathrm{R}}(k,k')e^{i\theta_k}\frac{\varepsilon_k\cos(\varepsilon_k t)\cos(\varepsilon_{k'}t)+\varepsilon_{k'}\sin(\varepsilon_k t)\sin(\varepsilon_{k'}t)}{\cos(k'-i\delta)-\cos(k+i\delta)}e^{i(k'm-kn)}$$
$$-\frac{i}{2}\int_{-\pi}^{\pi}\frac{dk}{2\pi}\int_{-\pi}^{\pi}\frac{dk'}{2\pi}G^{\mathrm{L}}(k,k')e^{i\theta_k}\frac{\varepsilon_k\cos(\varepsilon_k t)\cos(\varepsilon_{k'}t)+\varepsilon_{k'}\sin(\varepsilon_k t)\sin(\varepsilon_{k'}t)}{\cos(k'-i\delta)-\cos(k+i\delta)}e^{i(km-k'n)}e^{i(k'-k)}.$$
$$(165)$$

It is easy to check that relations (28) hold:

$$\langle A_n(t)A_m(t)\rangle = \langle B_{1-m}(t)B_{1-n}(t)\rangle \left(G^{\mathrm{L}}\leftrightarrow G^{\mathrm{R}}\right),\tag{166a}$$
$$\langle A_n(t)B_m(t)\rangle^{\mathrm{R}} = \langle A_{1-m}(t)B_{1-n}(t)\rangle^{\mathrm{L}}\left(G^{\mathrm{L}}\leftrightarrow G^{\mathrm{R}}\right).\tag{166b}$$

# C Some properties of the $g(k)$ and $I(k)$ functions

Here we analyze the integral expressions (155) and (158) and their derivatives at some special points. The resulting expressions can be used in the computation of the corrections to the semiclassical results (see Appendix D).

The first combination we consider is

$$g(\pi) - g(0) = I(\pi) - I(0) = \oint_{\mathscr{C}} \frac{dz}{\pi z} \frac{(1 - 2f_{q(z)})}{\sqrt{(h+z)(h+z^{-1})}} \,. \tag{167}$$

Since there are no poles any more at $|z| = 1$, we can rewrite this as ($z = e^{iq}$)

$$I(\pi) - I(0) = i \int_{-\pi}^{\pi} \frac{dq}{\pi} \frac{(1 - 2f_q)}{\varepsilon_q} = \frac{4i}{\pi} \int_{|1-h|}^{1+h} d\varepsilon \frac{1 - 2f(\varepsilon)}{\sqrt{(1+h)^2 - \varepsilon^2}\sqrt{\varepsilon^2 - (1-h)^2}} \,. \tag{168}$$

We note that for $f = 0$ ($T = 0$) the integral can be evaluated in closed form:

$$I(\pi) - I(0) = \frac{4i}{\pi|1-h|} K\left(\frac{-4h}{(1-h)^2}\right), \tag{169}$$

where $K(m)$ is the complete elliptic function of the first kind. If $f(\varepsilon) \neq 0$, the analytic properties of the integrand in (167) depends on the function $f(\varepsilon)$. For thermal distributions $f(\varepsilon) = 1/(1 + e^{\varepsilon/T})$, then $1 - 2f(\varepsilon) = \tanh[\varepsilon/(2T)]$ which is an odd function of $\varepsilon$ and this eliminates the branch cut coming from the denominator. In return the tanh function has poles on the real axis where $\varepsilon(z)/(2T) = ik\pi/2$ where $k$ is odd. The relevant solution of this equation is

$$z_k = -\frac{1 + h^2 + (k\pi T)^2 - \sqrt{[1 + h^2 + (k\pi T)^2]^2 - 4h^2}}{2h} \,. \tag{170}$$

Expanding the integrand around these points we find

$$\frac{1}{\pi z} \frac{\tanh[\sqrt{(h+z)(h+z^{-1})}/(2T)]}{\sqrt{(h+z)(h+z^{-1})}} = \frac{4T}{\pi \sqrt{[(k\pi T)^2 + (1-h)^2][(k\pi T)^2 + (1+h)^2]}} \frac{1}{z - z_k} + \dots, \tag{171}$$

so using the residue theorem we have to add the contributions of these infinitely many poles:

$$g(\pi) - g(0) = I(\pi) - I(0) = \sum_{k=1,k\,\text{odd}}^{\infty} \frac{8iT}{\sqrt{[(k\pi T)^2 + (1-h)^2][(k\pi T)^2 + (1+h)^2]}} \,. \tag{172}$$

Let us now investigate the derivative of $g(k)$. Differentiating the integrand of (155) we obtain

$$g'(k) = -2i \int_{-\pi}^{\pi} \frac{dq}{\pi} (1 - 2f_q^{\nu}) \frac{\sin^2 q}{4\varepsilon_q} \frac{\sin(k + i\delta)}{[\cos q - \cos(k + i\delta)]^2} \,. \tag{173}$$

This expression can be used to compute numerically $g'(k)$ for all $k$ except $k = 0, \pi$. At these points the integrand is not well-behaved (the cosine in the denominator becomes real). We need to get rid of the singularities, i.e. we need to start from Eq. (157). Here the integrand of $I'(k)$ contains a $\sin k$ factor (without $i\delta$) that vanishes at $k = 0, \pi$. We are left with the derivative of the first term in Eq. (157) which gives

$$g'(0) = \frac{1 - 2f_0}{1 + h}, \qquad g'(\pi) = -\frac{1 - 2f_\pi}{|1 - h|} \,. \tag{174}$$

We will also need the second derivatives at $k = 0$ and $k = \pi$. The second derivative of the first term in (157) vanishes at these momenta due to $\varepsilon'(0) = \varepsilon'(\pi) = 0$, so we are left with the second derivatives of $I(k)$. Differentiating the integrand in (158),

$$I''(0) = \oint_{\mathscr{C}} \frac{dz}{2\pi z} \frac{(1+z)^2}{(1-z)^2} \frac{(1-2f_{q(z)})}{\sqrt{(h+z)(h+z^{-1})}}, \tag{175}$$

$$I''(\pi) = -\oint_{\mathscr{C}} \frac{dz}{2\pi z} \frac{(1-z)^2}{(1+z)^2} \frac{(1-2f_{q(z)})}{\sqrt{(h+z)(h+z^{-1})}}. \tag{176}$$

Here the integrand still has a pole at $z = \pm 1$, so we cannot extend the contour to the unit circle in order to reintroduce $q$. For the Fermi–Dirac distribution $f(\varepsilon) = 1/(1 + e^{\varepsilon/T})$ we can again add the residues of the infinitely many simple poles of the tanh function to obtain

$$g''(0) = I''(0) = 4iT \sum_{k=1, k \text{ odd}}^{\infty} \frac{\sqrt{[(k\pi T)^2 + (1-h)^2][(k\pi T)^2 + (1+h)^2]}}{[(k\pi T)^2 + (1+h)^2]^2}, \tag{177}$$

$$g''(\pi) = I''(\pi) = -4iT \sum_{k=1, k \text{ odd}}^{\infty} \frac{\sqrt{[(k\pi T)^2 + (1-h)^2][(k\pi T)^2 + (1+h)^2]}}{[(k\pi T)^2 + (1-h)^2]^2}. \tag{178}$$

# D   Approach to the NESS

In this appendix we deal with the finite time corrections to the NESS, or in other words, with the late time approach of the steady state.

As a warm-up, let us focus on the profiles of the fermionic correlators near the origin. The Heaviside theta functions in Eqs. (36), (37) restrict the integrals on finite intervals set by $k_1$ and $k_2$ defined in Eq. (102). The NESS is given by $u = 0$ when $k_1 = -\pi, k_2 = 0$. For small $u$

$$k_1 \approx -\pi + \frac{|1-h|}{h} u, \qquad k_2 \approx -\frac{1+h}{h} u. \tag{179}$$

Then

$$\langle A_n(t)B_m(t)\rangle_{\text{sc}} =$$
$$-\int_{-\pi}^{k_1} \frac{dK}{2\pi} \cos(Kr + \theta_K)(1 - 2f_K^{\text{R}}) - \int_{k_2}^{\pi} \frac{dK}{2\pi} \cos(Kr + \theta_K)(1 - 2f_K^{\text{R}}) - \int_{k_1}^{k_2} \frac{dK}{2\pi} \cos(Kr + \theta_K)(1 - 2f_K^{\text{L}})$$
$$= \langle A_n(t)B_m(t)\rangle_{\text{NESS}} - \left(\int_{k_2}^{0} \frac{dK}{2\pi} + \int_{-\pi}^{k_1} \frac{dK}{2\pi}\right) \cos(Kr + \theta_K)[(1 - 2f_K^{\text{R}}) - (1 - 2f_K^{\text{L}})]$$
$$\approx \langle A_n(t)B_m(t)\rangle_{\text{NESS}} - \frac{1+h}{2\pi h} u \cos(\theta_0) 2(f_0^{\text{L}} - f_0^{\text{R}}) - \frac{|1-h|}{2\pi h} u \cos(r\pi + \theta_\pi) 2(f_{-\pi}^{\text{L}} - f_{-\pi}^{\text{R}})$$
$$= \langle A_n(t)B_m(t)\rangle_{\text{NESS}} - \frac{1+h}{\pi h} u(f_0^{\text{L}} - f_0^{\text{R}}) + (-1)^r \frac{1-h}{\pi h} u(f_\pi^{\text{L}} - f_\pi^{\text{R}}), \tag{180}$$

where we used that $\theta_0 = 0$ and $\theta_\pi = \pi$ for $h < 1$ and $\theta_\pi = 0$ for $h > 1$. The leading order correction gives a linear behavior near $u = 0$. For the correlators $\langle A_n(t)A_m(t)\rangle$ and $\langle B_n(t)B_m(t)\rangle$ the integrand is $\sin(rK)$ which vanishes both at $K = 0$ and $K = \pi$ thus the first order correction vanishes.

Now we analyze the behavior of the correlators in the limit $t \to \infty$, $n/t, m/t \to 0$. Following the method of Ref. [39], we take the derivative of our expressions with respect to time which results in the cancellation of the singular denominator.

### D.1 $\langle A_n(t)B_m(t)\rangle$

Let us start with the $\langle A_nB_m\rangle^{\text{R}}$ correlator. Differentiating Eq. (162) we get

$$\frac{\partial}{\partial t}\Big[\langle A_n(t)B_m(t)\rangle\Big]^{\text{R}} = -\frac{i}{2}\int_{-\pi}^{\pi}\frac{dk}{2\pi}\int_{-\pi}^{\pi}\frac{dk'}{2\pi}G^{\text{R}}(k,k')e^{i\theta_k}\frac{(\varepsilon_{k'}^2-\varepsilon_k^2)\sin(\varepsilon_k t)\cos(\varepsilon_{k'}t)}{\cos(k'-i\delta)-\cos(k+i\delta)}e^{i(k'm-kn)}. \tag{181}$$

Using $\varepsilon_{k'}^2-\varepsilon_k^2 = 2h[\cos(k')-\cos(k)]$ we find

$$\frac{\partial}{\partial t}\Big[\langle A_n(t)B_m(t)\rangle\Big]^{\text{R}} = -ih\int_{-\pi}^{\pi}\frac{dk}{2\pi}\int_{-\pi}^{\pi}\frac{dk'}{2\pi}G^{\text{R}}(k,k')e^{i\theta_k}\sin(\varepsilon_k t)\cos(\varepsilon_{k'}t)e^{i(k'm-kn)}. \tag{182}$$

By virtue of Eq. (159) the double integral factorizes:

$$\frac{i}{h}\frac{\partial}{\partial t}\Big[\langle A_n(t)B_m(t)\rangle\Big]^{\text{R}} = \int_{-\pi}^{\pi}\frac{dk}{2\pi}g^{\text{R}}(k)e^{i\theta_k}\sin(\varepsilon_k t)e^{-ikn}\cdot\int_{-\pi}^{\pi}\frac{dk'}{2\pi}\cos(\varepsilon_{k'}t)e^{ik'm}$$
$$-\int_{-\pi}^{\pi}\frac{dk}{2\pi}e^{i\theta_k}\sin(\varepsilon_k t)e^{-ikn}\cdot\int_{-\pi}^{\pi}\frac{dk'}{2\pi}g^{\text{R}}(k')\cos(\varepsilon_{k'}t)e^{-ik'm}, \tag{183}$$

where in the last integral we changed integration variable $k'\to -k'$.

All four integrals are now amenable to a standard stationary phase analysis. The stationary points $k=0$ and $k=\pi$ follow from $d\varepsilon_k/dt = 0$. The integrand is non-vanishing at these points so we have

$$\int_{-\pi}^{\pi}\frac{dk}{2\pi}F(k)\sin(\varepsilon_k t) = F(0)\sqrt{\frac{1+h}{2\pi h t}}\sin\Big((1+h)t-\frac{\pi}{4}\Big) + F(\pi)\sqrt{\frac{|1-h|}{2\pi h t}}\sin\Big(|1-h|t+\frac{\pi}{4}\Big)$$
$$-\frac{1}{2}F''(0)2\pi\Big(\frac{1+h}{2\pi h t}\Big)^{3/2}\cos\Big((1+h)t-\frac{\pi}{4}\Big) + \frac{1}{2}F''(\pi)2\pi\Big(\frac{|1-h|}{2\pi h t}\Big)^{3/2}\cos\Big(|1-h|t+\frac{\pi}{4}\Big) + \dots \tag{184}$$

and

$$\int_{-\pi}^{\pi}\frac{dk}{2\pi}F(k)\cos(\varepsilon_k t) = F(0)\sqrt{\frac{1+h}{2\pi h t}}\cos\Big((1+h)t-\frac{\pi}{4}\Big) + F(\pi)\sqrt{\frac{|1-h|}{2\pi h t}}\cos\Big(|1-h|t+\frac{\pi}{4}\Big) +$$
$$\frac{1}{2}F''(0)2\pi\Big(\frac{1+h}{2\pi h t}\Big)^{3/2}\sin\Big((1+h)t-\frac{\pi}{4}\Big) - \frac{1}{2}F''(\pi)2\pi\Big(\frac{|1-h|}{2\pi h t}\Big)^{3/2}\sin\Big(|1-h|t+\frac{\pi}{4}\Big) + \dots, \tag{185}$$

where the dots stand for terms of order $\mathcal{O}(t^{-5/2})$. Substituting the various integrands for the function $F(k)$, expanding the products and collecting the terms we find, using $\theta(0)=0$,

$$\frac{\partial}{\partial t}\Big[\langle A_n(t)B_m(t)\rangle\Big]^{\text{R}} = -i\frac{\sqrt{|1-h^2|}}{2\pi t}[g(0)-g(\pi)]\Big[(-1)^m\sin\Big((1+h)t-\frac{\pi}{4}\Big)\cos\Big(|1-h|t+\frac{\pi}{4}\Big)$$
$$-(-1)^n e^{i\theta(\pi)}\cos\Big((1+h)t-\frac{\pi}{4}\Big)\sin\Big(|1-h|t+\frac{\pi}{4}\Big)\Big]$$
$$+\frac{1}{4\pi h t^2}\Big[(1+h)^2(g'(0)(m+n-\theta'(0)+ig''(0))$$
$$-(-1)^{m+n}e^{i\theta(\pi)}(1-h)^2(g'(\pi)(m+n-\theta'(\pi)+ig''(\pi))\Big] + \frac{\text{osc.}}{t^2} + \mathcal{O}(t^{-3}), \tag{186}$$

where osc./$t^2$ stands for terms of order $t^{-2}$ which are oscillating. After integration with respect to $t$, these will remain in the same order giving subleading contribution.[10] However, non-oscillating $\mathcal{O}(t^{-2})$ terms need to be separated as they will become $\mathcal{O}(t^{-1})$ after integration.

---

[10] $\int dt\,\sin(at)/t^2 \approx -\cos(at)/(at^2)$ and $\int dt\,\cos(at)/t^2 \approx -a\pi/2 + \sin(at)/(at^2)$.

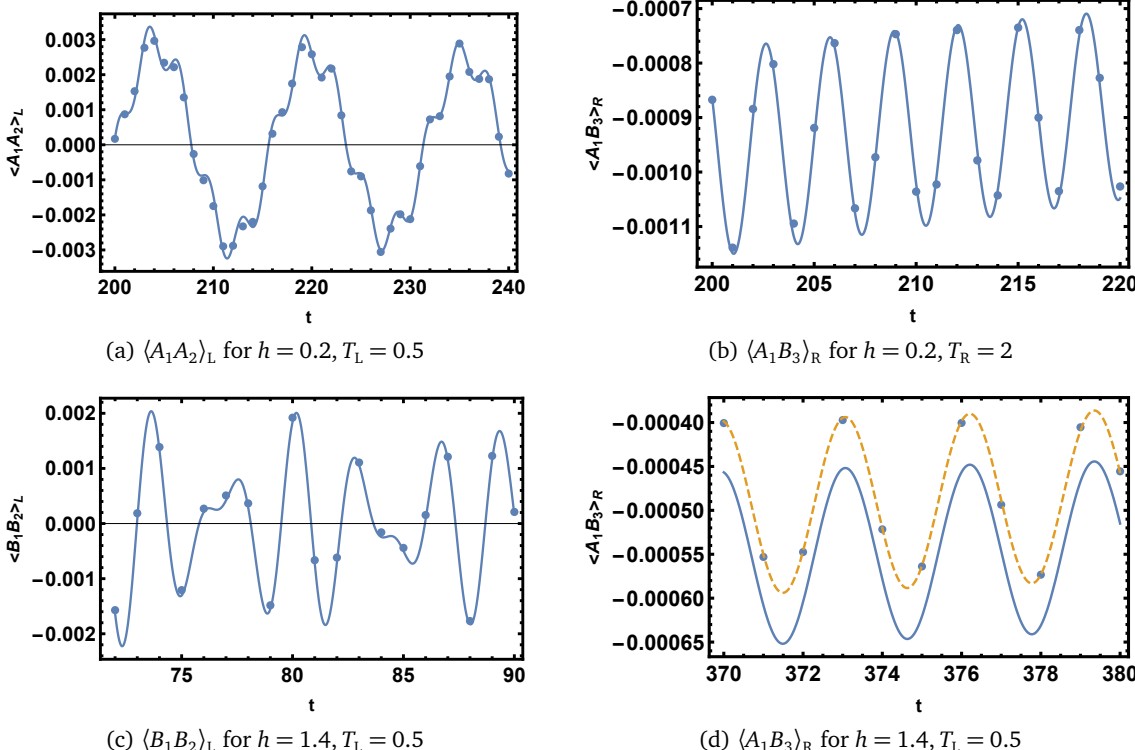

(a) $\langle A_1 A_2 \rangle_\text{L}$ for $h = 0.2, T_\text{L} = 0.5$

(b) $\langle A_1 B_3 \rangle_\text{R}$ for $h = 0.2, T_\text{R} = 2$

(c) $\langle B_1 B_2 \rangle_\text{L}$ for $h = 1.4, T_\text{L} = 0.5$

(d) $\langle A_1 B_3 \rangle_\text{R}$ for $h = 1.4, T_\text{L} = 0.5$

Figure 7: Fermionic correlations near the junction as functions of time. Then numerical data (dots) are plotted together with the analytic results (42),(46) (solid line) that include the leading $\mathcal{O}(1/t)$ corrections as well. In (d) an additional constant shift is needed for a perfect agreement, which is a subleading $\mathcal{O}(1/t^2)$ correction.

At the evaluation of (186) the results of Appendix C can be used. We note that

$$\theta'(0) = \frac{1}{1+h}, \qquad \theta'(\pi) = \frac{1}{1-h}, \tag{187}$$

and $\theta(\pi) = 0$ for $h > 1$ and $\theta(\pi) = \pi$ for $h < 1$. We can rewrite the first term as

$$i \frac{\sqrt{|1-h^2|}}{2\pi} [g(0) - g(\pi)] \left[ \frac{(-1)^m + \text{sgn}(h-1)(-1)^n}{2} \frac{\cos[2\min(1,h)t]}{t} \right. $$
$$\left. - \frac{(-1)^m - \text{sgn}(h-1)(-1)^n}{2} \frac{\sin[2\max(1,h)t]}{t} \right]. \tag{188}$$

Now we can integrate the result to obtain $\langle A_n(t) B_m(t) \rangle$. Using $\int \mathrm{d}t \, \sin(at)/t \approx \pi/2 - \cos(at)/(at)$ and $\int \mathrm{d}t \, \cos(at)/t \approx \sin(at)/(at)$ for large $t$, we obtain Eq. (42) of the main text. The correction to $\langle A_n(t) B_m(t) \rangle^\text{L}$ can be obtained using Eq. (28).

We compare the analytic expressions with numerical data in Fig. 7. In most cases we find excellent agreement. In some cases, however, an additional constant shift is necessary to get agreement. We checked that this shift is proportional to $1/t^2$ to a very good precision, thus it is a subleading correction not contained in our formula.

### D.2 $\langle A_n(t)A_m(t)\rangle$

Differentiating Eq. (160),

$$
\frac{\partial}{\partial t}\Big[\langle A_n(t)A_m(t)\rangle\Big]^{\mathrm{R}} =
$$
$$
-\frac{1}{2}\int_{-\pi}^{\pi}\frac{\mathrm{d}k}{2\pi}\int_{-\pi}^{\pi}\frac{\mathrm{d}k'}{2\pi}G^{\mathrm{R}}(k,k')e^{i(\theta_k-\theta_{k'})}\frac{(\varepsilon_{k'}^2-\varepsilon_k^2)\sin(\varepsilon_k t)\sin(\varepsilon_{k'}t)}{\cos(k'-i\delta)-\cos(k+i\delta)}e^{i(k'm-kn)}. \quad (189)
$$

Using $\varepsilon_{k'}^2-\varepsilon_k^2=2h[\cos(k')-\cos(k)]$ we find

$$
-\frac{1}{h}\frac{\partial}{\partial t}\Big[\langle A_n(t)A_m(t)\rangle\Big]^{\mathrm{R}} = \int_{-\pi}^{\pi}\frac{\mathrm{d}k}{2\pi}g^{\mathrm{R}}(k)e^{i\theta_k}\sin(\varepsilon_k t)e^{-ikn}\cdot\int_{-\pi}^{\pi}\frac{\mathrm{d}k'}{2\pi}e^{i\theta_{k'}}\sin(\varepsilon_{k'}t)e^{-ik'm}-
$$
$$
-\int_{-\pi}^{\pi}\frac{\mathrm{d}k}{2\pi}e^{i\theta_k}\sin(\varepsilon_k t)e^{-ikn}\cdot\int_{-\pi}^{\pi}\frac{\mathrm{d}k'}{2\pi}g^{\mathrm{R}}(k')e^{i\theta_{k'}}\sin(\varepsilon_{k'}t)e^{-ik'm}. \quad (190)
$$

Substituting the various integrands for the function $F(k)$, expanding the products and collecting the terms we find, using $\theta(0)=0$,

$$
\frac{\partial}{\partial t}\Big[\langle A_n(t)A_m(t)\rangle\Big]^{\mathrm{R}} = [(-1)^m-(-1)^n]\frac{\sqrt{|1-h^2|}}{4\pi t}[g^{\mathrm{R}}(0)-g^{\mathrm{R}}(\pi)]\mathrm{sgn}(1-h)\times
$$
$$
\Big(\sin[2\min(1,h)t]-\cos[2\max(1,h)t]\Big)+\frac{\mathrm{osc.}}{t^2}+\dots \quad (191)
$$

Unlike $\langle A_n B_m\rangle$, in the $t^{-2}$ order there are only oscillating terms that are subleading after integration. Now we can integrate the result to obtain Eq. (46a) of the main text. Formula (46b) for $\langle A_n(t)A_m(t)\rangle^{\mathrm{L}}$ is obtained in an analogous fashion. Using the relation Eq. (28) we readily obtain the corresponding expressions for $\langle B_n(t)B_m(t)\rangle$. These results are also compared with numerics in Fig. 7.

## E  Edge behavior of the fermionic correlations

In Sec. 5.2 we derived the universal scaling form of the front of the fermionic correlations in terms of the Airy kernel. However, comparison with the numerical simulations at finite time shows large deviations between the scaling form and the numerical data indicating that the leading order corrections are important.

The calculation of these corrections are explained in Sec. 5.2. For clarity, let us spell out in a bit more detail the derivatives of the function $F(k,q)$ in Eq. (58). Using $\theta'_\kappa=\theta'_{-\kappa}=1$,

$$
\partial_1 F(\kappa,\kappa)=\frac{1}{2}G(\kappa,\kappa)e^{i\theta_\kappa}\frac{\varepsilon_\kappa}{2}\left(\frac{g'(\kappa)}{G(\kappa,\kappa)}+\frac{\varepsilon'_\kappa}{2\varepsilon_\kappa}\right), \quad (192)
$$

$$
\partial_2 F(\kappa,\kappa)=\frac{1}{2}G(\kappa,\kappa)e^{i\theta_\kappa}\frac{\varepsilon_\kappa}{2}\left(\frac{g'(-\kappa)}{G(\kappa,\kappa)}+i+\frac{\varepsilon'_\kappa}{2\varepsilon_\kappa}\right). \quad (193)
$$

The derivative of $g(k)$ can be computed from Eq. (173).

In Fig. 8 we compare the numerical data with the analytic results with and without the corrections. In all the cases taking into account the corrections yields significantly better agreement.



Figure 8: Edge behavior of the fermionic correlations for $h = 0.2, T_L = 0.5, T_R = 2, t = 900$. Numerical results (dots) are compared with the scaling forms (dashed) and with the analytic results incorporating the first order corrections (solid line).

## F   Correlation length via the Szegő lemma

We would like to simplify expression (91) for the correlation length. Using $\Theta(x)^2 = \Theta(x)$, $\Theta(-x) = 1 - \Theta(x)$ we find

$$
\begin{aligned}
-F(k)^2 + G(k)G(-k) = \\
(1-2f_k^L)(1-2f_k^R) + 2(1-2f_k^R)(f_k^L - f_k^R)\Theta(v_k + u) - 2(1-2f_k^L)(f_k^L - f_k^R)\Theta(v_k - u) \\
-4(f_k^L - f_k^R)^2\Theta(v_k - u)\Theta(v_k + u). \quad (194)
\end{aligned}
$$

Let us distinguish the cases $u \geq 0$ and $u < 0$. When $u \geq 0$, $\Theta(v_k - u)\Theta(v_k + u) = \Theta(v_k - u)$, while for $u < 0$, $\Theta(v_k - u)\Theta(v_k + u) = \Theta(v_k + u)$, so

$$
\begin{aligned}
-F(k)^2 + G(k)G(-k) = \\
\Theta(u)\left\{(1 - 2f_k^{\text{L}})(1 - 2f_k^{\text{R}}) + 2(1 - 2f_k^{\text{R}})(f_k^{\text{L}} - f_k^{\text{R}})[\Theta(v_k + u) - \Theta(v_k - u)]\right\} + \\
\Theta(-u)\left\{(1 - 2f_k^{\text{L}})(1 - 2f_k^{\text{R}}) + 2(1 - 2f_k^{\text{L}})(f_k^{\text{L}} - f_k^{\text{R}})[\Theta(v_k + u) - \Theta(v_k - u)]\right\}. \quad (195)
\end{aligned}
$$

Then we can split the domain of integation in (91) into three parts as

$$
\begin{aligned}
\xi^{-1} = -\Theta(u)\frac{1}{2}\int_{-\pi}^{\pi}\frac{\mathrm{d}k}{2\pi}\{[\Theta(-v_k - u) + \Theta(v_k - u)]\ln\left[(1 - 2f_k^{\text{L}})(1 - 2f_k^{\text{R}})\right] \\
+ 2\Theta(u - v_k)\Theta(u + v_k)\ln(1 - 2f_k^{\text{R}})\} \\
-\Theta(-u)\frac{1}{2}\int_{-\pi}^{\pi}\frac{\mathrm{d}k}{2\pi}\{[\Theta(u - v_k) + \Theta(u + v_k)]\ln\left[(1 - 2f_k^{\text{L}})(1 - 2f_k^{\text{R}})\right] \\
+ 2\Theta(v_k - u)\Theta(-u - v_k)\ln(1 - 2f_k^{\text{L}})\}. \quad (196)
\end{aligned}
$$

In each integral, the two step functions in the first term give equal contributions due to the symmetry of the integrand. In the second terms we use again identity $\Theta(-x) = 1 - \Theta(x)$ for one of the step functions. These manipulations lead to identical integrands in the two lines, and we arrive at Eq. (92).

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
