# Peer review of "Inhomogeneous quenches in the transverse field Ising chain: scaling and front dynamics"

_SciPost Physics, doi:SciPost Phys. 3, 020 (2017)_

## Round 1 · Referee Report · Anonymous · 2017-5-4

Strengths

1-timely topic (quantum quenches starting from inhomogeneous initial state)
2-very detailed derivations and comparison between numerical evaluation of exact expressions and semi-classical approximations

Weaknesses

1-maybe one could add a link to the general results of Ref. 20

Report

The article studies quantum quenches starting from an inhomogeneous initial state in the transverse-field Ising chain. All relevant observables are analysed in terms of the exact solution as well as a semi-classical approximate solution. The presentation is very detailed and well written.

Requested changes

1-At the beginning of Sec. 4 the semi-classical limit is defined as $t\to\infty$ while $n/t,m/t$ is kept fixed and $(n-m)/t\to 0$. I do not see how $(n-m)/t\to 0$ if $n/t,m/t$ are fixed. Please clarify this.
2-In Sec. 6.2 a discussion of the relation to the general results of Ref. 20 could be added.

  • validity: high
  • significance: good
  • originality: good
  • clarity: high
  • formatting: perfect
  • grammar: perfect

Author:  Márton Kormos  on 2017-07-28  [id 158]

(in reply to Report 1 on 2017-05-04)

I thank the Referee for reading the manuscript, for their very positive evaluation, and for drawing my attention to the following issues.

"1-At the beginning of Sec. 4 the semi-classical limit is defined as
$t\to\infty$ while n/t,m/t is kept fixed and $(n-m)/t\to0$. I do not see how (n-m)/t\to0 if $n/t,m/t$ are fixed. Please clarify this."

I have replaced the quoted sentence by
"This semiclassical or hydrodynamic limit corresponds to the case when $n,m,t\to\infty$ with $\lim n/t=\lim m/t$ fixed implying $(n-m)/t\to0$."

"2-In Sec. 6.2 a discussion of the relation to the general results of Ref. 20 could be added."

Ref. [20] focuses on the full statistics of the energy current in the NESS in conformal field theories. It has already been shown in Ref. [16] that the critical Ising spin chain reproduces the general CFT results of Ref. [20]. I have added a reference to [20] below Eq. (67).

---

## Round 1 · Referee Report · Anonymous · 2017-6-5

Strengths

1- The problem studied is very interesting;
2- The paper finds exact results and controlled expansions;

Weaknesses

1- Not enough discussion of the results;
2-The bound state for h<1 is not considered;

Report

The paper studies nonequilibrium dynamics in the transverse field Ising chain (TFIC), focussing on inhomogeneous initial states that give rise to nontrivial transport. Specifically, the initial states considered are the junction of two homogeneous thermal states at different temperatures and the "domain wall" state.

The author maps the transverse field Ising chain in a free fermionic model and derives exact double integral representations for the two point fermionic correlators, with fermions at generic positions $n$ and $m$. Expanding this result for large $n, m$ and time he finds that the leading contribution gives the so called ''semi-classical" or "hydrodynamic" limit, which can be obtained by using the semiclassical picture of moving quasi-particles. He also determines the first finite time corrections to the NESS fermionic correlations (i.e. he considers the limit where the positions of the fermionic operators are fixed and the time goes to infinity), finding that they are $O(1/t)$. Finally, he determines the leading contribution to the correlations at the edge of the propagating front ($n\sim m \sim v_{\rm max} t$, where $v_{\rm max}$ is the maximal propagation velocity of quasiparticles), finding that they are described by the Airy kernel, he also determines the first corrections. He uses these results to compute profiles of observables (such as density and current of energy and magnetization) and correlation functions.

I think that the paper is interesting. Exact results and controlled expansions in free models are very important to test the validity of general ideas, such as the hydrodynamic approach. Moreover, the author also finds explicitly some of the finite time corrections, showing for example how the non equilibrium steady state is approached. Even if the paper is quite technical, the author made a substantial effort to make it more readable, moving most of the technical parts to the appendices. I, therefore, recommend the publication of this paper in Scipost.

I have two main points that I think should be addressed before publication, one more general and the other more technical.

1) I think that the only weak point of the paper is the too hasty discussion of the results obtained. As I detail in the specific points later in a number of cases the author could examine a little bit more the results obtained, discussing their importance and their dependence on the parameters, the relation with other works, and why one should expect those results on based on some general principles.

2) After Eq (11) the author writes that for $\frac{L}{L+1}<h<1$ there is a complex solution for the quantization conditions, but it does not affect the results in the thermodynamic limit. This is incorrect, as it follows from what correctly reported in Appendix A, the complex solution appears for $h<\frac{L}{L+1}$. This means that the complex solution remains in the thermodynamic limit for $h<1$. Since the bound state is localised at $x=0$ its contribution will not affect the results in the scaling limit for finte $u$. In the $u=0$ case it could in principle affect observables close to the origin. I think that the author should analyse the contribution of this bound state.

Requested changes

I have the following comments

- In the introduction the author writes that integrable model have " an extensive number of locally conserved quantities". I think that this sentence should read as "an extensive number of local conserved quantities". It is indeed believed that also local defects can break the integrability.

- Even if, as the author says, a thorough analysis of the partitioning protocol in the TFIC was missing (the work: Perfetto and Gambassi arXiv:1704.03437 is contemporary to the current one), a tightly related problem was considered last year in: Bertini and Fagotti, Physical Review Letters 117, 130402 (2016). This paper presents the leading order expansion for large $n$, $m$ and time of all the fermionic correlators in a quench from a generic gaussian translationally invariant state to the TFIC with open boundary conditions. Also in that case, it was found that the result is described by the semiclassical approximation. I think that the author should compare to this work.

- In the introduction, when talking about the TFIC the author refers to [18], saying that in there the energy current in the NESS and its fluctuations are computed. This is probably a typo as the model studied in there is not the TFIC.

- Can the author say something about the corrections away from the NESS? Does he expect also those to be $O(1/t)$?

- Is there a reason why one expects the Airy kernel to describe the scaling near the front in general?

- After Eq. (56) the author could note that what he finds is in agreement with Lieb-Robinson bounds.

- In Figures 1 and 2 the energy density is called $u_n$ while in the text $h_n$.

- Does the author have a physical interpretation for purely quantum the energy backflow phenomenon observed?

- I am surprised by how large are the corrections to the leading order result (68) (note that the equation referenced in the caption is incorrect) in Fig. 4 (b) and how well agree with the numerics. When the corrections are larger than the result I would expect the time to be too short for the expansion (53) and (58) to apply (this seems not to be the case given the agreement with the numerics).

- In Equation (73) would be useful to remind the reader of the definition of $u$, as that quantity is defined much before in the text.

- I think that would be better to study the critical regime for $T_L=0$ and $T_R=\infty$ separately, not in the "Tranverse Magnetization" Section (note that in the inline equation after 74 the author uses $u_n$ for the energy density).

- In Section 6.5 I think that the author should at least include the footnote 6 in the main text, i.e. should explicitly write that he restricts to $h<1$.

- The typo "naiv" appears two times, once in Section 6.5 and once in the Conclusion.

- In the last line of Eq. (174) $u$ is replaced by $x$.

  • validity: high
  • significance: high
  • originality: good
  • clarity: high
  • formatting: excellent
  • grammar: good

Author:  Márton Kormos  on 2017-07-28  [id 160]

(in reply to Report 2 on 2017-06-05)

I would like to thank the Referee for the very careful, critical reading of the manuscript and for the insightful comments and questions. I must say that the paper has definitely improved upon implementing the suggested modifications.

"1) I think that the only weak point of the paper is the too hasty discussion of the results obtained. As I detail in the specific points later in a number of cases the author could examine a little bit more the results obtained, discussing their importance and their dependence on the parameters, the relation with other works, and why one should expect those results on based on some general principles."

In the revised version I have made an effort to improve this aspect of the manuscript. There are more discussions of the results throughout the paper following the comments of all the referees. There is a new subsection 5.2.2 on the edge behavior in the critical case. I have extended the conclusions section. Following the comments of Referee 3, I have also included comments on the lattice effect shown by the leading finite time corrections. Please find my answers to the specific questions below.

"2) After Eq (11) the author writes that for $h<1$ there is a complex solution for the quantization conditions, but it does not affect the results in the thermodynamic limit. This is incorrect, as it follows from what correctly reported in Appendix A, the complex solution appears for $h<1$. This means that the complex solution remains in the thermodynamic limit for $h<1$. Since the bound state is localised at $x=0$ its contribution will not affect the results in the scaling limit for finite $u$. In the $u=0$ case it could in principle affect observables close to the origin. I think that the author should analyse the contribution of this bound state."

All the numerical results shown in the plots were obtained using Eqs. (19), (25), and (26), where in the summation over the modes the boundary bound states were also included for $h<1.$ Their effect is however negligible, as demonstrated by the excellent agreement between the numerical and analytic results shown in several plots. Even the leading finite time corrections to the NESS are seen to agree with the numerics (see Figs. 1, 3, and 7).

To exclude the logical possibility that there is an error in the formulas or in their numerical implementation, I have compared the short time behavior of correlations near the junction with exact diagonalization results and found perfect agreement.

The reason for the unimportance of the boundary state is that it can only affect the dynamics through the initial correlations. The $AA$ and $BB$ type correlations give a Kronecker delta in the presence or absence of the boundary state, while the contribution of the boundary state to $\langle A_jB_l \rangle_0$ is negligible. This follows from Eqs. (25b,c) in which $\phi$ and $\psi$ are localized at opposite edges of the chain so their overlap is exponentially small.

Let me turn to the list of requested changes.
I have corrected the typos and implemented the small changes requested in comments number 1, 3, 6, 7, 10, 11, 12, 13.

"- Even if, as the author says, a thorough analysis of the partitioning protocol in the TFIC was missing (the work: Perfetto and Gambassi arXiv:1704.03437 is contemporary to the current one), a tightly related problem was considered last year in: Bertini and Fagotti, Physical Review Letters 117, 130402 (2016). This paper presents the leading order expansion for large $n$, $m$ and time of all the fermionic correlators in a quench from a generic gaussian translationally invariant state to the TFIC with open boundary conditions. Also in that case, it was found that the result is described by the semiclassical approximation. I think that the author should compare to this work."

I thank the Referee for reminding me of this nice work. I have inserted references to this paper at several places, including the Introduction and Sec. 4 (in the first paragraph and below Eq. (37) and below Eq. (39)).

"- Can the author say something about the corrections away from the NESS? Does he expect also those to be $O(1/t)$?"

A derivation very similar to that in Appendix D could also be performed at a ray corresponding to a finite $n/t\approx m/t$ value. The stationary point would be shifted in this case, of course. The natural expectation is that the leading order corrections are also $O(1/t),$ which is consistent with the numerical checks I have performed. I have added a remark about this at the end of Sec. 5.1.

"- Is there a reason why one expects the Airy kernel to describe the scaling near the front in general?"

The mathematical origin of the Airy function near the front is that at the stationary point of the phase of the integrand the second derivative of the phase vanishes. If this happens in the double integral whose integrand has a pole when both integration variables approach this degenerate critical point, the Airy kernel appears. This is thus a rather generic phenomenon in fermionic systems. I have added a paragraph with references below Eq. (55).

However, not all correlations are described by the Airy kernel. In the revised version I point out that the correlator $\langle A_nB_n\rangle$ related to the the transverse magnetization and Majorana fermion density does not follow the Airy kernel behavior in the ferromagnetic phase. Moreover, in the critical case the derivative of the Airy kernel describes all correlations which does not show the typical staircase structure.

"- Does the author have a physical interpretation for purely quantum the energy backflow phenomenon observed?"

As Fig. 4b shows, the backflow phenomenon is captured by the correction at the front described by the Airy kernel and its derivatives. Based on this it would be interesting to make a connection with the Airy process or some generalization of it related to the motion of the rightmost particle, but this was beyond the scope of the paper.

"- I am surprised by how large are the corrections to the leading order result (68) (note that the equation referenced in the caption is incorrect) in Fig. 4 (b) and how well agree with the numerics. When the corrections are larger than the result I would expect the time to be too short for the expansion (53) and (58) to apply (this seems not to be the case given the agreement with the numerics)."

I completely agree with the Referee and I have to admit that I do not have a good explanation for this. However, as also noted by the Referee, the numerics fully support the analytic calculations.

"- I think that would be better to study the critical regime for $T_L=0$ and $T_R=\infty$ separately, not in the "Tranverse Magnetization" Section (note that in the inline equation after 74 the author uses u_n for the energy density)."

I agree that it is not optimal to quote results for the energy density and current in the transverse magnetization section. However, as the discussion of this very special case is hardly longer than a paragraph, it would not be logical to devote a whole (sub)section to it, this is why I decided to include it in another section.

"- In the last line of Eq. (174) $u$ is replaced by $x$."

$u$ was correctly replaced by $x/t,$ but this is an unnecessary complication so I decided to keep simply $u$.

---

## Round 1 · Referee Report · Anonymous · 2017-6-6

Strengths

1- Interesting and timely subject
2- Self-contained work
3- The analytical predictions are checked against numerics

Weaknesses

1- Long and somehow technical paper
2- The relevance of the new results is unclear

Report

This paper studies time evolution following an inhomogeneous quench in the transverse-field Ising chain (TFIC). Specifically, two open chains are prepared in equilibrium at different temperatures; then, the coupling between two spins at the boundaries is turned on, and the system is let to unitarily time evolve. As also mentioned by the author, this protocol of nonequilibrium dynamics has attracted much attention, and it was already analytically investigated in many integrable models, including the TFIC. I think it is fair to say that the new result of this paper is to show the leading finite-time corrections to the non-equilibium steady state (NESS) emerging in the limit of infinite time. To my knowledge, such finite-time corrections have never been discussed before, so I think that this work gives a substantial contribution to the field of research.
In my opinion the paper is written rather well, but it looks like a technical paper. For example, the author did not include a summary of the results, and the reader is forced to read the entire paper to become aware of all the findings. In addition, there are natural questions that remain unanswered, like: "Why are the finite-time corrections important?", "What do I learn from the new findings?". I recommend this paper for publication, however I think that the author should improve these aspects.

Requested changes

1- I suggest that the author include a section where the new results are presented and discussed.
2- It is not clear to me if the author has taken into account the contribution from the bound state when h<1.
3- At the end of section 4, the author writes: "In our approach, the NESS appears as a single special member of a continuous family that gives the space-time profile of correlations in the semiclassical limit". The author should consider to add a reference to [Bertini and Fagotti, Phys. Rev. Lett. 117, 130402 (2016)], which, to the best of my knowledge, is the first paper stressing that point.
4- The leading finite-time corrections to the NESS display parity effects. I think that the author should comment on this aspect, especially in relation to the recent attempts to go beyond the hydrodynamic description assuming that the leading corrections can still be described within a local density approximation.
5- The expressions for the fermion correlations close to the light cone (eqs (53) and (55)) seem to approach zero in the infinite-time limit. Is this correct? Shouldn't some correlations approach a finite value?
6-Do the leading finite-time corrections to the NESS depend on the details of the initial state close to the junction? Is there something universal in the leading finite-time corrections?

  • validity: top
  • significance: high
  • originality: good
  • clarity: high
  • formatting: perfect
  • grammar: excellent

Author:  Márton Kormos  on 2017-07-28  [id 159]

(in reply to Report 3 on 2017-06-06)

I would like to thank the Referee for the very careful, critical reading of the manuscript and for the insightful comments and questions. I must say that the paper has definitely improved upon implementing the suggested modifications.

"I think it is fair to say that the new result of this paper is to show the leading finite-time corrections to the non-equilibium steady state (NESS) emerging in the limit of infinite time."

The finite time corrections go beyond the semiclassical or hydrodynamical approximation so they are hard to analyze in general. The Ising model gives us the rare opportunity to derive the leading corrections analytically. For example, this can provide hints about the applicability of hydrodynamic approaches to describe these corrections.

I consider the derivation of the fine structure of the front equally important. On the one hand it reveals that the Airy kernel also shows up in the Ising spin chain, on the other hand I found that, perhaps surprisingly, the fermion density (transverse magnetization) is not described by the Airy kernel. Moreover, in the critical case none of the correlations follow the Airy kernel.

Finally, the extensive numerical simulations also revealed some interesting details about the dynamics and showed that a naive application of the Szeg\H{o} lemma describes the longitudinal spin-spin correlations remarkably well.

"1- I suggest that the author include a section where the new results are presented and discussed.
"Why are the finite-time corrections important?", "What do I learn from the new findings?"

I seriously considered this suggestion. However, in the end I decided not to add another section but to extend the Conclusions, adding references to the relevant figures and equations. Besides this, there are more discussions of the results throughout the paper following the comments of all the referees. I have added a new subsection about the front structure in the critical case.

"2- It is not clear to me if the author has taken into account the contribution from the bound state when $h<1$. "

All the numerical results shown in the plots were obtained using Eqs. (19), (23), and (26), where in the summation over the modes the boundary bound states were also included for $h<1.$ Their effect is however negligible, as demonstrated by the excellent agreement between the numerical and analytic results shown in several plots.

For a more detailed explanation see my answer to the first question of Referee 1.

"3- At the end of section 4, the author writes: "In our approach, the NESS appears as a single special member of a continuous family that gives the space-time profile of correlations in the semiclassical limit". The author should consider to add a reference to [Bertini and Fagotti, Phys. Rev. Lett. 117, 130402 (2016)], which, to the best of my knowledge, is the first paper stressing that point."

I thank the Referee for reminding me of this nice piece of work. I have inserted references to this paper at several places, including the Introduction and Sec. 4 (in the first paragraph and below Eq. (37) and below Eq. (39)).

"4- The leading finite-time corrections to the NESS display parity effects. I think that the author should comment on this aspect, especially in relation to the recent attempts to go beyond the hydrodynamic description assuming that the leading corrections can still be described within a local density approximation."

I would like to thank the Referee for bringing up this important point. I have included a paragraph at the end of Sec. 5.1 about this observation, and mentioned it in the Abstract and the Conclusions.

"5- The expressions for the fermion correlations close to the light cone (eqs (53) and (55)) seem to approach zero in the infinite-time limit. Is this correct? Shouldn't some correlations approach a finite value?"

This comment of the Referee was very valuable as it led to an improved discussion of the calculation of the near edge behavior of correlation functions. For correlations that are non-zero to the right of the front, one first has to subtract this finite value. It turns out that this expectation value is related to the residue of the corresponding double integral, so subtracting it amounts to changing the integration contour in the integral (i.e. to changing the sign of the infinitesimal shift $i\delta$.) The resulting double integral then yields the Airy kernel that decays to zero outside the front.

Let me note that in the literature usually the left contributions are analyzed only which decay to zero, and to the best of my knowledge the subtraction step has not been discussed.

"6-Do the leading finite-time corrections to the NESS depend on the details of the initial state close to the junction? Is there something universal in the leading finite-time corrections?"

The leading finite time corrections to the NESS depend on the initial state only through the function $g(p)$ and its derivatives at momenta $p=0$ and $p=\pi.$ The function $g(p)$ is determined by the initial distribution $f(p)$. So different distributions that give the same values $g^{(n)}(p)$ ($n=0,1,2$) will give rise to the same leading corrections, but I cannot make a better or deeper statement at present.

---

## Round 2 · Referee Report · Anonymous · 2017-8-2

Report

The author has carefully considered all the remarks raised by the referees and revised the manuscript accordingly.

---

## Round 2 · Referee Report · Anonymous · 2017-8-19

Report

The author addressed the majority of the points I raised and substantially improved the paper. Before publication, however, I think that the author should mention Ref.[75] when discussing the critical case in the newly introduced Section 5.2.2 and in all the passages of Sec. 6 and 8 where he mentions the critical case. Indeed, it is true that the first version of the current paper appeared at the same time as Ref. [75] but in that version there was no mention to the critical case, which was instead thoroughly studied in Ref. [75].

I also suggest to address the following additional points.

Requested changes

1) The condition for the appearance of bound states mentioned after eq 11 is still wrong. As I mentioned in my previous report, the correct one is $h<L/(L+1)$. Regarding the effects of boundary bound states on physical observables, my opinion is that the contribution is negligile because of the particular form (eq. 25) of the initial states considered in this work.

2) In the revised Section 5.2 (and in his response) the author says that the correlator $<A_nB_n>$ is related to the density of Majorana fermions. What is the density of Majornana fermions? I think that the author meant the density of Jordan-Wigner fermions (those described by the operators $c_n$ (cf. eq 2)).

3) At the end of the newly introduced Section 5.2.2 the author writes "... the scaling is $\sim t^{-2/3}$ instead of $\sim t^{-1/3}$". I think that the word scaling here is misleading, the width of the region is still scaling as $t^{1/3}$. I suggest to write something on the lines of "decays with a different power of t".

4) At the beginning of the Conclusion when describing the initial states considered I suggest to mention the condition in eq. 23, i.e. the zero anomalous correlation request. This would also make clearer the final part of the Conclusion when the author discusses the possible generalizations.

  • validity: high
  • significance: high
  • originality: good
  • clarity: high
  • formatting: excellent
  • grammar: good

Author:  Márton Kormos  on 2017-08-23  [id 164]

(in reply to Report 2 on 2017-08-19)

I thank the Referee for his/her second report. I am happy that in his/her opinion I have "substantially improved the paper".

I considered all the comments and requests of the Referee and modified the manuscript accordingly. In particular, I cite Ref. [75] in Sec. 5.2.2, at the end of Sec. 6.2 and in Sec. 8.

1) The condition for the appearance of bound states mentioned after eq 11 is still wrong. As I mentioned in my previous report, the correct one is $h<L/(L+1)$. Regarding the effects of boundary bound states on physical observables, my opinion is that the contribution is negligile because of the particular form (eq. 25) of the initial states considered in this work.

I agree with the Referee. I have corrected the condition and added a footnote on page 6: "The edge modes can be more important for other initial states."

2) In the revised Section 5.2 (and in his response) the author says that the correlator $\langle A_nB_n\rangle$ is related to the density of Majorana fermions. What is the density of Majornana fermions? I think that the author meant the density of Jordan-Wigner fermions (those described by the operators $c_n$ (cf. eq 2)).

Indeed, this is what I meant. I corrected the sentence in Sec. 5.2.

3) At the end of the newly introduced Section 5.2.2 the author writes "... the scaling is $\sim t^{−2/3}$ instead of $\sim t^{−1/3}$". I think that the word scaling here is misleading, the width of the region is still scaling as $\sim t^{1/3}$. I suggest to write something on the lines of "decays with a different power of t".

I have changed the quoted sentence.

4) At the beginning of the Conclusion when describing the initial states considered I suggest to mention the condition in eq. 23, i.e. the zero anomalous correlation request. This would also make clearer the final part of the Conclusion when the author discusses the possible generalizations.

I thank the Referee for this suggestion, I have added a sentence about it at the beginning of the Conclusion.

---

## Round 2 · Author Response

Dear Editor,

I submit the revised version of the manuscript, following the suggestions of the referees who found my work "interesting" and of high significance. All the referees made valid and very useful comments and asked relevant questions. I believe I managed to answer all the questions and I have implemented the requested changes. In particular, I extended the discussions of the results at several places. Thanks to the remarks of the referees the paper has definitely improved.

As all three referees recommended the publication of the manuscript after minor revision, I hope that the new version meets all the criteria for publication in SciPost.

Yours sincerely,
Marton Kormos

---

## Round 2 · List of Changes

List of changes (typos or minor changes are not included):

Changed the abstract, introduction, and conclusion. References to main results (equations and figures) included in the Conclusions to improve readability.

Added refs. [42,47,48,68,69,70,72,75].

Connection made with Ref. [42] in he Introduction and Sec. 4 (in the first paragraph, below Eq. (37) and below Eq. (39)).

Remark added at the end of Sec. 5.1. about the leading corrections at finite $x/t$.

Remark added at the end of Sec. 5.1. and in the Conclusions about the possible implications of the lattice effect showing up in the leading corrections to the NESS.

Paragraph added below Eq. (55) about the universal nature of the Airy kernel.

Paragraph added below Eq. (56) discussing the correlations that are not described by the Airy kernel near the front (e.g. the fermion density in the ferromagnetic phase).

Changed the derivation of the front structure in Sec 5.2 to correctly take into account the finite asymptotic value of correlations outside the light cone.

Added subsection 5.2.2. on the edge behavior in the critical case.

Added footnote 7 making it clear that the boundary edge mode was included in the numerical calculation for $h<1$.

Changed Fig. 1a and 2b to show the critical case.

---

## Round 3 · Author Response

Dear Editor,

I resubmit my manuscript in which I have implemented all the minor changes requested by Referee 2.

Kind regards,
Marton Kormos

---

## Round 3 · List of Changes

1. Ref. [75] is now cited in Sec. 5.2.2, at the end of Sec. 6.2 and in Sec. 8 related to the edge behavior in the critical case.

  2. The condition for the appearance of bound states after Eq. (11) is corrected and a footnote is added.

  3. In Sec. 5.2 "density of Majorana fermions" is replaced by "density of Jordan-Wigner fermions".

  4. A sentence regarding the critical edge scaling in the last paragraph of Sec. 5.2.2 is changed.

  5. The requirement of vanishing anomalous correlations in the initial state is mentioned at the beginning of the Conclusions.

---

## Editorial Decision

published